# A deep proteomics perspective on CRM1-mediated nuclear export and nucleocytoplasmic partitioning

Koray Kırlı[1†], Samir Karaca[1,2†], Heinz Jürgen Dehne[1], Matthias Samwer[1‡], Kuan Ting Pan[2], Christof Lenz[2,3], Henning Urlaub[2,3*], Dirk Görlich[1*]

[1]Department of Cellular Logistics, Max Planck Institute for Biophysical Chemistry, Göttingen, Germany; [2]Bioanalytical Mass Spectrometry Group, Max Planck Institute for Biophysical Chemistry, Göttingen, Germany; [3]Bioanalytics, Institute for Clinical Chemistry, University Medical Center Göttingen, Göttingen, Germany

**Abstract** CRM1 is a highly conserved, RanGTPase-driven exportin that carries proteins and RNPs from the nucleus to the cytoplasm. We now explored the cargo-spectrum of CRM1 in depth and identified surprisingly large numbers, namely >700 export substrates from the yeast *S. cerevisiae*, ≈1000 from *Xenopus* oocytes and >1050 from human cells. In addition, we quantified the partitioning of ≈5000 unique proteins between nucleus and cytoplasm of *Xenopus* oocytes. The data suggest new CRM1 functions in spatial control of vesicle coat-assembly, centrosomes, autophagy, peroxisome biogenesis, cytoskeleton, ribosome maturation, translation, mRNA degradation, and more generally in precluding a potentially detrimental action of cytoplasmic pathways within the nuclear interior. There are also numerous new instances where CRM1 appears to act in regulatory circuits. Altogether, our dataset allows unprecedented insights into the nucleocytoplasmic organisation of eukaryotic cells, into the contributions of an exceedingly promiscuous exportin and it provides a new basis for NES prediction.

*For correspondence: henning.urlaub@mpibpc.mpg.de (HU); goerlich@mpibpc.mpg.de (DG)

[†]These authors contributed equally to this work

Present address: [‡]Institute of Molecular Biotechnology of the Austrian Academy of Sciences, Vienna, Austria

Competing interests: The authors declare that no competing interests exist.

## Introduction

The nuclear envelope (NE) separates the cell nucleus from the cytoplasm. Although its lipid bilayers are impermeable for macromolecules, embedded nuclear pore complexes (NPCs) allow an exchange of material between these compartments (*Feldherr, 1962*). The NPC permeability barrier controls this exchange. It grants small molecules a free passage, but becomes increasingly restrictive as the size of the mobile species approaches or exceeds a diameter of ≈ 5 nm (*Mohr et al., 2009*). Shuttling nuclear transport receptors (NTRs) are not bound by this restriction (for review see: *Kimura and Imamoto, 2014*). They can traverse NPCs by facilitated translocation and have the capacity to ferry even large cargoes, such as newly assembled ribosomal subunits, across the barrier.

Active transport of cargoes against concentration gradients requires an intact NE and NPC-barrier for retaining already transported cargoes in the destination compartment. In addition, it requires an input of metabolic energy, typically by the RanGTPase system. The corresponding duty cycles include not only a switch of Ran between its GDP- and GTP-bound states, but also one round of Ran-shuttling between nucleus and cytoplasm. The energetic coupling occurs through a primary RanGTP-gradient, which is generated through NTF2-mediated import of RanGDP, nucleotide exchange by the nuclear RanGEF (RCC1) and cytoplasmic RanGTP-depletion by RanGAP and RanBP1/RanBP2. This gradient then directly fuels importin- and exportin-mediated transport cycles.

Exportins (reviewed in *Güttler and Görlich, 2011*) bind their cargo molecules cooperatively with RanGTP inside nuclei, carry them as trimeric RanGTP·exportin·cargo complexes to the cytoplasm,

**eLife digest** Animals, plants and other eukaryotic organisms subdivide their cells into compartments that carry out specific tasks. For example, the cell nucleus hosts the genome and handles the genetic information, whereas the surrounding cytoplasm is specialized in making proteins. These proteins are then either used in the cytoplasm or transported to the nucleus and other cell compartments. Since proteins are not made in the nucleus, all proteins in this compartment must be imported from the cytoplasm.

Two layers of membrane separate the nucleus and cytoplasm from each other. Any exchange of material must therefore proceed through channels called nuclear pore complexes, or NPCs for short. The NPCs have filters that allow only small molecules a free transit, while larger ones are typically rejected. However, larger proteins may also rapidly pass through the nuclear pore complexes if loaded onto dedicated shuttle molecules; for example, "exportins" transport proteins out of the nucleus.

Kırlı, Karaca et al. used an approach called proteomics to measure the levels of 5,000 different proteins within the nucleus and the cytoplasm. Such a census helps to predict where a given protein works and where it might cause problems. Further experiments also used proteomics to identify which proteins are carried by an exportin called CRM1. This revealed that a remarkably large number of different proteins (around 1,000) are exported by CRM1 from either yeast, human or frog cell nuclei. Most of these "cargo" proteins were previously thought to never leave the cytoplasm. It now seems, however, that these proteins can leak into the nucleus, but CRM1 recognizes them as cytoplasmic proteins and expels them from the nucleus.

These findings suggest that the border control at NPCs is less perfect than was previously believed. If not remedied, this would pose a serious problem for the cell, because the accumulation of "wrong" proteins inside the nucleus would disturb the processes that occur there and could destabilize the genome. Kırlı, Karaca et al. propose that the export of such accidentally displaced proteins by CRM1 is a crucial measure to protect the nucleus.

where GTP-hydrolysis triggers release of cargo and Ran. The free exportin can then re-enter nuclei and export the next cargo.

CRM1, also called exportin 1 or Xpo1, is the most conserved nuclear export receptor (*Adachi and Yanagida, 1989*; *Fornerod et al., 1997a*; *Fukuda et al., 1997*; *Stade et al., 1997*). It is structurally well characterised, whereby structures of free CRM1, of certain cargo•CRM1•RanGTP complexes as well as their assembly- and disassembly-intermediates have been solved (*Dong et al., 2009*; *Monecke et al., 2009*; *Güttler et al., 2010*; *Koyama and Matsuura, 2010*; *Monecke et al., 2013*; *Saito and Matsuura, 2013*; *Koyama et al., 2014*).

CRM1 recognises its cargoes through short linear nuclear export signals (NESs), which comprise 4–5 critical hydrophobic (Φ) residues with characteristic spacings (*Wen et al., 1994*; *Fischer et al., 1995*; *Dong et al., 2009*; *Monecke et al., 2009*; *Güttler et al., 2010*). CRM1 is essential for viability and is the target of the potentially lethal bacterial toxin leptomycin B (*Nishi et al., 1994*), which blocks NES-binding by covalently modifying a conserved cysteine residue within the NES-binding site (*Kudo et al., 1999*; *Dong et al., 2009*; *Monecke et al., 2009*; *Sun et al., 2013*). CRM1 is known to carry multiple cargoes, including newly assembled 40S and 60S ribosomal subunits, the signal recognition particle SRP, U snRNAs, or the genomic RNA of HIV1 via the Rev protein (*Moy and Silver, 1999*; *Ciufo and Brown, 2000*; *Ho et al., 2000*; *Ohno et al., 2000*; *Gadal et al., 2001*; *Thomas and Kutay, 2003*; *Rouquette et al., 2005*). It is also known to regulate key cellular events by conditional export of transcription factors or cell cycle regulators from the nucleus (see e.g. *Hagting et al., 1998*; *Yan et al., 1998*; *Yang et al., 1998*).

Cell nucleus and cytoplasm are prime examples for the division of labour in a eukaryotic cell. The cytoplasm hosts the machineries of the secretory pathway, many metabolic activities as well as the cytoskeletal structures that account for cell motility and long-range transport. It has also specialised in protein synthesis and *de novo* protein folding.

The nucleus lacks protein synthesis and thus depends on protein import from the cytoplasm. It has specialised in DNA replication and repair, nucleosome assembly, transcription, ribosome assembly, as well as in mRNA splicing and processing. Such specialisation critically relies on a spatial separation of interfering activities: Intranuclear protein synthesis, for example, would be a particularly wasteful process, because ribosomes would then also translate unspliced or incompletely spliced mRNAs, consequently read into introns, add inappropriate residues to the nascent chains, eventually encounter premature stop codons and thus produce truncated protein fragments. Such aberrant translation products would not only be non-functional, but probably also toxic, because they fail to fold, or act in a dominant-negative fashion.

It is thus not very surprising that eukaryotic cells have implemented several lines of defence against intranuclear translation, whereby the NE acts as a primary barrier to keep cytoplasmic translation activity out of nuclei. Likewise, even though the 40S and 60S ribosomal subunits assemble inside the nucleus, they gain full translation competence only following late maturation steps in the cytoplasm (reviewed in *Panse and Johnson, 2010*; *Thomson et al., 2013*).

A very general problem is, however, that the NPC barrier is not perfect. Instead, also objects larger than the nominal exclusion limit can leak—albeit slowly—into the nucleus (*Bonner, 1975*; *Mohr et al., 2009*). Such slow mixing of nuclear and cytoplasmic contents would become a problem if the leaked-in proteins would interfere with nuclear functions or dysregulate cellular activities. Countermeasures might be selective degradation or inhibition in the inappropriate compartment, or, when mis-localised to the nucleus, recognition by an exportin and subsequent retrieval to the cytoplasm.

Indeed, precedents for such exportin-mediated back-sorting of cytoplasmic proteins from the nucleus are already known. Animal Xpo6, for example, keeps actin out of the nucleus (*Stüven et al., 2003*), while Xpo4 and Xpo5 do the same for the translation elongation factors eIF5a (*Lipowsky et al., 2000*) and eEF1A respectively (*Bohnsack et al., 2002*; *Calado et al., 2002*). CRM1 was shown to expel several cytoplasmic factors from the nuclear compartment, including the RanGTPase system components RanBP1 (*Plafker and Macara, 2000*) and RanGAP (*Feng et al., 1999*) as well as the translation factor subunits eIF2β, eIF5, eIF2Bε and eRF1 (*Bohnsack et al., 2002*). The full extent of active cytoplasmic confinement has, however, not yet been assessed.

We report here global scale analyses of nucleocytoplasmic partitioning in *Xenopus* oocytes and of CRM1-mediated nuclear export. According to stringent criteria, we identified ≈ 1000 potential CRM1 cargoes from *Xenopus laevis* oocytes, ≈ 1050, from human HeLa cells, and ≈ 700 from the yeast *S. cerevisiae*. We tested a subset of cargo candidates for CRM1-dependent nuclear export in cultured human cells and thereby validated a majority of them as true cargoes. For a subset, we also confirmed direct CRM1-interaction and mapped the corresponding NESs, some of which turned out to have unusual features. The majority of identified CRM1 cargoes are proteins and protein complexes with a very strong bias towards a cytoplasmic localisation, suggesting that their active back-sorting from the nucleus is a major cellular activity. This applies to nearly all translation factors (including the largest translation factor complexes like eIF2B), to factors involved in final ribosomal maturation steps, which might prevent ribosomes from acquiring translation competence already in the nuclear compartment, as well as to regulatory proteins, autophagy-linked factors, peroxisome biogenesis factors and to centrosomal proteins, in both, humans and frogs. Another major functional group of CRM1 cargoes with perfect nuclear exclusion is represented by vesicle coat proteins, which points to a strong evolutionary pressure to preclude the budding of vesicles from the inner nuclear membrane. We also identified numerous new instances, where CRM1 appears to act in regulatory circuits. More generally, these data represent a very rich resource for other researchers seeking information about nucleocytoplasmic distribution and CRM1-controlled localisation.

## Results and discussion

### Nucleocytoplasmic protein partitioning in *Xenopus laevis* oocytes

We were interested in a global view of how soluble proteins and protein complexes partition between the nucleus and the cytoplasm. In order to tackle this question, we applied a deep proteome analysis to the isolated compartments. A problem for such endeavour is that standard cell fractionation procedures rely on shearing forces, often combined with hypotonic lysis or even treatment

with detergents (see e.g. *Blobel and Potter, 1966*; *Dignam et al., 1983*). All these treatments compromise the integrity of the NE. Nuclear proteins, which are not firmly associated with solid structures like chromatin, will then leak out and contaminate the cytoplasmic fraction—just as the nuclear fraction will be contaminated by cytoplasmic components.

In order to avoid these problems, we turned to *Xenopus laevis* stage VI oocytes (*Dumont, 1972*). These cells measure ≈1.3 mm in diameter and have nuclei of ≈450 µm. Such very large dimensions allow for a manual oocyte dissection into nuclear and cytoplasmic fractions with exceptionally little cross-contamination (see e.g. *De Robertis et al., 1978*). These oocyte nuclei are also special with their volume being 100,000 times larger than that of average-sized cells with a G2 DNA contents. The chromatin should therefore make no more than a negligible contribution to nuclear retention of proteins. Instead, the nucleocytoplasmic distribution of a given protein or protein complex in these cells should be solely determined by its passive diffusion properties and by their potential to access active nuclear import and/or export pathways. In addition, oocytes are very long-lived cells that grow over months to their final size, which implies that even slow partitioning processes are likely to have reached a steady state.

As a standard experiment, we dissected 60 oocytes, cleared the cytoplasmic fractions off yolk, normalised the nuclear and cytoplasmic fractions to their respective volumes, and identified proteins in three biological replicates by mass spectrometry. Proteins of two replicates were separated by SDS-PAGE (*Figure 1A and 1B*) and in-gel digested with trypsin. As a complementary approach, proteins of the third replicate were digested in solution. Resulting peptides were separated by reverse phase chromatography at pH 10 and obtained fractions analysed by LC-MS/MS. The raw data were

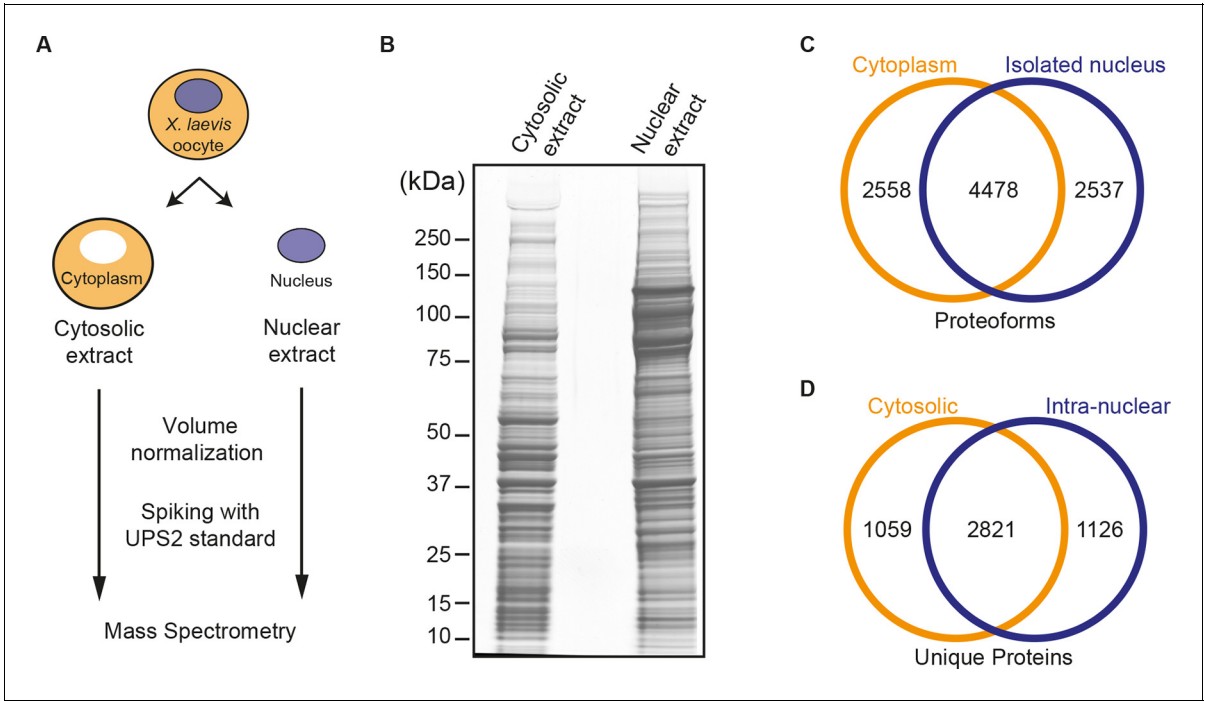

**Figure 1.** Spatial proteomics of *Xenopus laevis* oocytes. (A) Workflow for mass spectrometric analysis of cytosolic and nuclear proteins. For details, see Materials and methods and main text. (B) Analysis of obtained cytosolic and nuclear fractions by SDS-PAGE and Coomassie-staining. The loads correspond to 750 nanolitres of either yolk-free cytoplasm or nuclear contents. (C) Venn diagram of proteoforms (including all allelic variants of a given gene product) that have been identified in the manually isolated cytoplasms and nuclei. (D) Venn diagram is similar to (C), but proteoforms corresponding to a given gene (foremost allelic variants) have been merged down to 'unique proteins'. Also, proteins were subtracted that just co-purified with nuclei, but do not represent intranuclear proteins; this applied to constituents of the nuclear envelope (ER) and the nucleus-associated mitochondrial cloud.

The following figure supplement is available for figure 1:

**Figure supplement 1.** Estimation for the accuracy of mass spectrometric protein quantitation.

searched against a comprehensive *Xenopus laevis* database (*Wühr et al., 2014*). In this way, a total of 9573 proteoforms (*Smith et al., 2013*) were identified, 7015 in isolated nuclei and 7036 in the cytoplasmic fractions. The intersection set comprised 4478 proteoforms (*Figure 1C*).

*Xenopus laevis* is pseudotetraploid; thus it shows greater allelic diversity than other species (*Hellsten et al., 2007*), which represents a challenge for peptide-based protein quantification. We therefore treated allelic isoforms not as separate proteins, but mapped all recognizable allelic forms down to unique protein species. This 'mapping' was guided also by comparisons to the human and the (diploid) *Xenopus tropicalis* proteomes (see Materials and methods).

We aimed at a high-quality dataset for those proteins that can actually pass through NPCs and partition between the nucleus and cytoplasm. We therefore tried to minimize the number of 'contaminants' in the list, which represent, for example, the endoplasmic reticulum (which co-purifies with nuclei in the form of the NE) or the mitochondrial cloud (*Heasman et al., 1984*) that is tightly associated with the outside of these nuclei. We subsequently tested which proteins disappeared from the nuclear fraction when the NE was manually removed. Such proteins were discarded from the list if their sequence features qualified them also as integral membrane, ER-luminal or mitochondrial proteins. This left us with a total of 5006 unique proteins. Of these, 1126 were identified only in the nuclear fraction, 1059 only in cytosolic fraction and 2821 in both compartments (*Figure 1D*, and *Supplementary file 1*).

In order to quantify each of these proteins in the nuclear and cytoplasmic compartments, we employed the iBAQ strategy (*Schwanhäusser et al., 2011*) combined with an internal universal protein standard (UPS2; see Materials and methods). We estimate that our quantitation was reliable over a range of 5 orders of magnitude in abundance and accurate within a factor of 2.3 (see *Figure 1—figure supplement 1*).

To validate the quality of the obtained nuclear and cytoplasmic fractions, we first evaluated the partitioning of the previously described marker proteins nucleophosmin (also called B23 or NO38) and gelsolin, which are localised exclusively to nucleus (*Schmidt-Zachmann et al., 1987*) or cytoplasm (*Yin and Stossel, 1979*; *Samwer et al., 2013*), respectively. We chose highly abundant proteins, because these yield a larger number of unique, quantifiable peptides and thus allow for a more precise quantification. For nucleophosmin (a histone chaperone), we measured a nuclear concentration of ≈ 2.3 µM, a cytoplasmic concentration of 8 nM, and hence a nucleocytoplasmic (N:C) partition coefficient of 300:1. For gelsolin (a factor that stabilizes cytoplasmic actin in its G-form) we measured a nuclear concentration of ≈ 0.01 µM, a cytoplasmic of ≈ 4 µM, and thus an N:C ratio of 1:400. This suggests that the obtained nuclear and cytoplasmic fractions show only very limited cross-contamination and also suggested what range of partition coefficients should be expected also for other proteins.

*Supplementary file 1* contains complete and simplified data sets for the nucleocytoplasmic distribution of 5006 individual proteins and 9573 proteoforms. It lists the measured concentrations in nucleus and cytoplasm as well as ratios on a log10 scale. To make the data as accessible as possible to other researchers, we included not only unique identifiers for each protein hit, but also clickable annotation links to the corresponding UniProt entries for the identified *Xenopus* proteins (X. *laevis* if available, otherwise *X. tropicalis*) as well as to the human orthologues, where annotations as of now are more complete.

When broken down to functional groups, it becomes evident that the various cellular processes represent quite characteristic N:C distribution patterns. As these patterns are in many cases tightly linked to the activity of CRM1, we will discuss these after describing our approach of mapping the CRM1-dependent nuclear exportome.

## Implications for energy supply of the oocyte nucleus

An apparently CRM1-independent aspect relates to the energy supply of the giant oocyte cell nucleus. Efficient duty cycles of numerous enzymes, such as Ran, require a high NTP:NDP ratio, which seems hard to maintain inside these nuclei if ATP were produced only outside (i.e. by mitochondria or cytosolic glycolysis). The problem arises from the short half-life of ATP in living cells (≈ 1 min), which is less time than an ATP molecule would typically need to diffuse into and across such large nucleus. We now found evidence in the oocyte for two parallel solutions to this problem.

The first is a 'creatine-creatine phosphate energy shuttle', which uses diffusion of creatine phosphate (CP) instead of ATP for a long-range transport of energy equivalents (reviewed by

*Bessman and Carpenter, 1985*). It exploits the ≈160 times higher phosphorylation potential of CP as compared to ATP, and operates by creatine kinase (CK) synthesising CP from ATP near ATP sources and the reverse reaction at sites of ATP consumption. Oocytes enable such CP shuttle by maintaining high CK concentrations in both, cytoplasm and nucleus (3 µM and 1.5 µM, respectively; *Supplementary file 1*).

The second solution concerns the glycolysis pathway. The oocyte nucleus contains all glycolytic enzymes, except for hexokinase and phosphofructokinase that catalyse the ATP-consuming preparatory steps. The hexokinase level is generally very low in oocytes (because they produce glucose 6-phosphate by other means; see *Nutt et al., 2005* and *Supplementary file 1*), while 6-phosphofructokinase (PFK) is confined to the cytoplasm (N:C ≈ 1:2000; *Supplementary file 1*). This enzyme distribution predicts a directed flux of fructose 1,6 bisphosphate into the nucleus. It further suggests that oocyte nuclei produce the high-energy compounds phosphoenolpyruvate and 1,3 bisphoglycerate locally and use them for synthesizing ATP and other NTPs.

## A large-scale data set for potential CRM1 cargoes from three species

The exportin CRM1 is known to be essential for viability and to account for nuclear export of numerous targets (see e.g. *Xu et al., 2012b*; *Thakar et al., 2013* for a comprehensive summary of so far identified substrates). However, it has been unclear how broad the cargo spectrum really is and which set of cellular processes are directly or indirectly controlled by CRM1-dependent nuclear export.

In order to close this gap, we set out to identify cargoes in an unbiased manner from three different species and types of cells, namely: the already-mentioned *Xenopus laevis* oocytes, human HeLa cells and the yeast *S. cerevisiae*. We first prepared cellular extracts, used immobilised CRM1 as an affinity matrix and asked which proteins and protein complexes would bind in a RanGTP-dependent manner. As detailed below, CRM1 has many interaction partners of widely different abundance and affinity, which implies that the majority of them bind to the immobilized exportin in a highly sub-stoichiometric manner. The challenge thus is to cleanly distinguish such highly sub-stoichiometric bands from non-specific interactors. To this end, we optimised several parameters, such as the way of immobilisation, the exportin:extract ratios (to minimise competition between cargoes), buffer conditions, incubation time, etc., in order to maximise the signal-to-noise ratio of the affinity chromatography (for details see Materials and methods).

*Figure 2A* shows the starting extracts and CRM1-bound fractions of such affinity chromatographies, and it documents indeed a very large number of protein species that bound to CRM1 in a RanGTP-dependent manner. Using the mass-spectrometric approaches mentioned above, we identified in the starting extracts ≈2800 (*Xenopus*), ≈3900 (human) and ≈2600 (yeast) proteins (*Figure 2B*). In the 'CRM1+RanGTP' samples, ≈2300, 3000 and ≈2000 proteins were identified. About 14% of these had not been detected in the total extracts and these probably represent low-abundance proteins that had been highly enriched during CRM1 affinity chromatography.

The lists of proteins identified in the CRM1+RanGTP-bound fractions include not only true CRM1 binders and CRM1 export cargoes, but, for sure, also false-positive ones. To classify a given hit as a promising candidate, we therefore relied not only on its mere identification in this fraction. We also considered to which extent it became enriched from the input extract ('input enrichment'), to which extent its CRM1-binding had been stimulated by RanGTP ('RanGTP-stimulation'), as well as its absolute abundance in the 'CRM1+RanGTP'-bound fraction ('Molar fractions in CRM1+RanGTP') (*Figure 3*), which affects the accuracy of quantification and thus the reliability of the first two numbers. As a resource for other researchers, we organised the quantitative data in Excel sheets (*Supplementary files 2–4*), which contain not only the just mentioned numbers, but also clickable links to the corresponding UniProt entries, as well as cellular localisations derived either from databases (human and yeast) or measured directly (*Xenopus*).

We divided the identified proteins into several categories, using as a criterion species-specific thresholds (*Supplementary files 2–4*, and *Figure 3*). These thresholds had been adjusted to best match the behaviour of proteins, whose specific CRM1-interaction is already established beyond doubt, as well as of proteins that are known not to interact with the exportin (see Materials and methods). CRM1-binders of the category A1 not only had to pass the most stringent thresholds in terms of RanGTP-stimulation of CRM1-binding (≥500-fold in the case of *Xenopus*) and enrichment from the input extract (≥3-fold), but also belong to the most abundant proteins in the 'CRM1

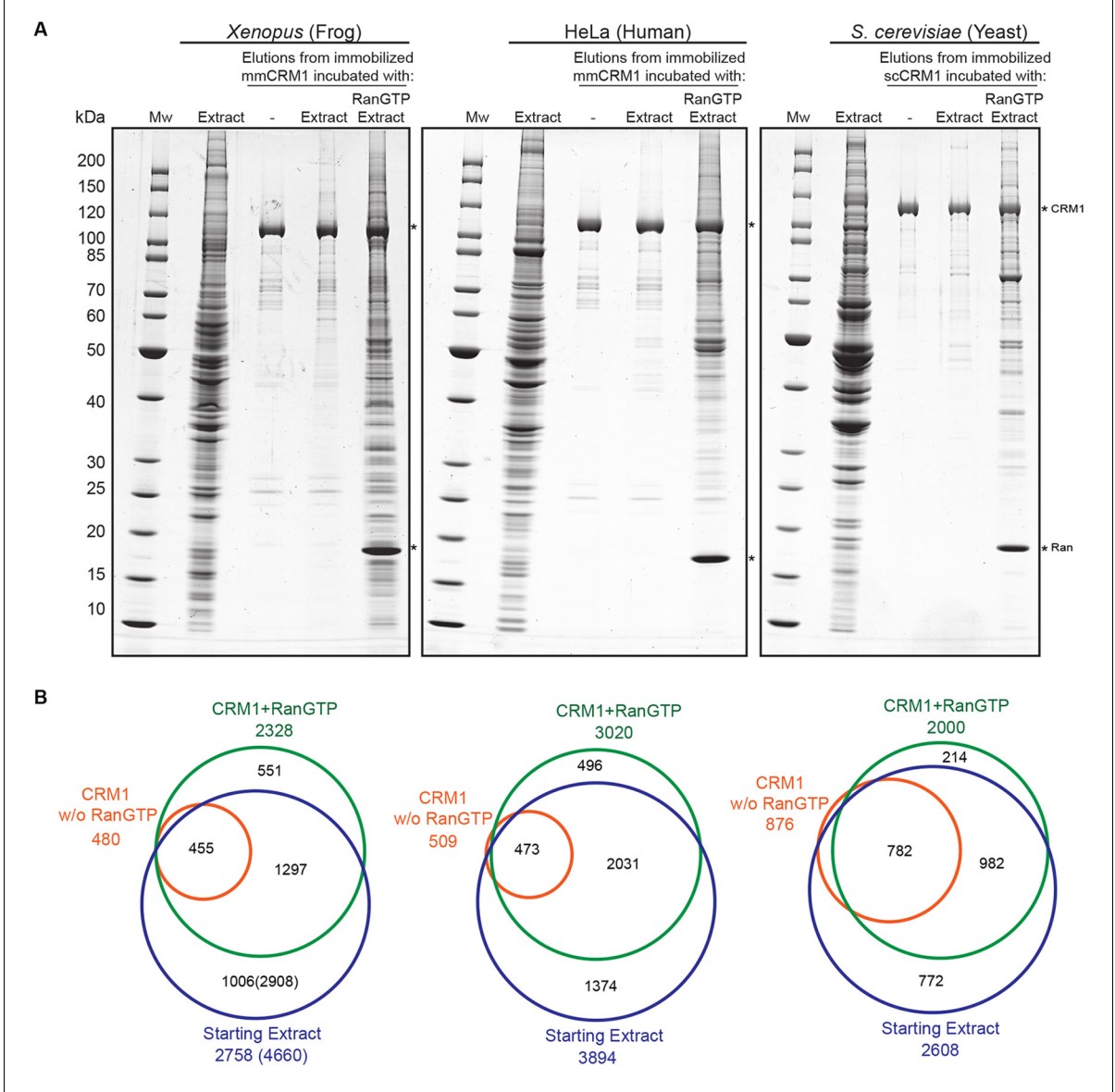

**Figure 2.** Identification of potential CRM1 cargoes from 3 species. (**A**) Mouse (mm) or yeast (sc) CRM1 were immobilised through a biotinylated Avi-tag to streptavidin agarose, and incubated with indicated extracts (1 ml), without or with the addition of 5 µM RanQ69L$^{5-180}$GTP. The beads were thoroughly washed and subsequently eluted at 45°C with SDS sample buffer (which leaves the biotin-streptavidin interaction largely intact). Analysis of indicated samples was by SDS-PAGE and Coomassie-staining. 1/200 of the starting extracts and 1/5 of eluates were loaded. (**B**) Starting extracts, CRM1 w/o Ran, and CRM1+RanGTP samples were analysed by mass spectrometry. Venn diagrams represent identified unique proteins. Numbers in parenthesis include also proteins that were not identified in a total *Xenopus* extract or the 'CRM1+RanGTP' sample, but in the isolated nuclear and cytoplasmic fractions; these proteins extend the list of 'CRM1-non-binders'.

+RanGTP'-bound fraction (Note that the threshold of a 500 times stronger signal than in the negative (minus RanGTP) control is far more stringent than the standard of '2-fold', which is used in most proteomics studies). A2 had an even stricter threshold for 'input enrichment' (≥100), but a relaxed one for the apparent RanGTP-stimulation. It includes cargoes like snurportin, which bind CRM1 so strongly that the affinity is still high even in the absence of RanGTP (*Paraskeva et al., 1999*). In category B, one of the three criteria was relaxed, while the category 'low abundant' includes proteins that were only detected in the 'CRM1+RanGTP'-bound fraction, but were not sufficiently abundant to qualify for category A.

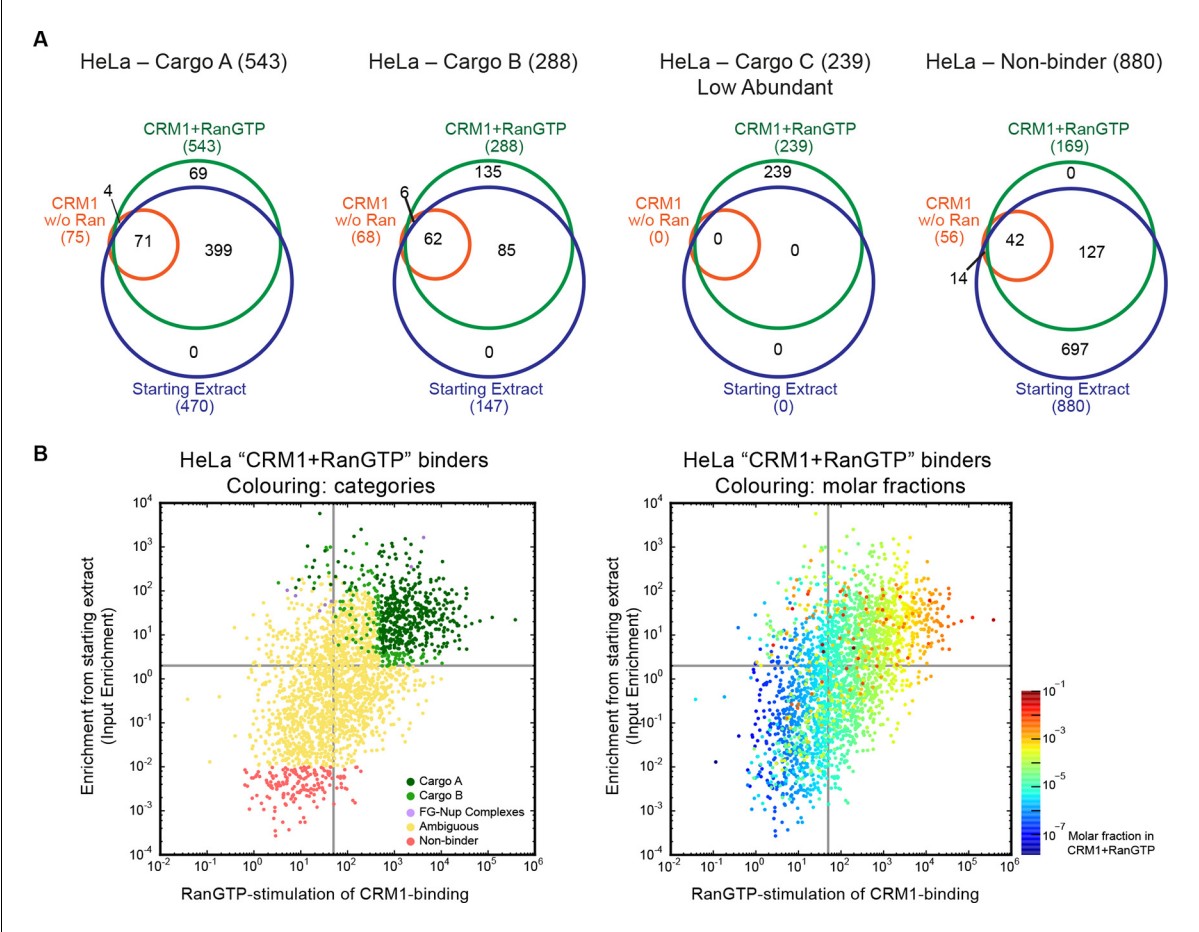

**Figure 3.** Categories of CRM1-binders from HeLa cells. For each identified CRM1-binder, we calculated or estimated three parameters from measured iBAQ intensities: its abundance (molar fraction) within the 'CRM1+RanGTP'-bound sample, the RanGTP-stimulation of its CRM1-binding, and how strongly it became enriched by the 'CRM1+RanGTP'-affinity chromatography. These numbers where then used to group binders into distinct categories, ranging from 'A' (the most probable cargoes) to 'non-binders'. (A) Venn diagrams representing the indicated cargo classes with respect to their identification in the starting extract, 'CRM1 w/o Ran'- and/or 'CRM1+RanGTP'-bound samples. (B) Scatter plot representing 'CRM1+RanGTP'-binders from HeLa cells, using the parameters 'RanGTP-stimulation' and 'input-enrichment' as coordinates. Colouring is according to classification. Most 'non-binders' had not been identified in the 'CRM1+RanGTP' sample; they are therefore also not plotted. Measurement of the parameters 'RanGTP-stimulation' and 'input-enrichment' required the identification a given candidate in input, 'CRM1 w/o Ran', as well as in the 'CRM1+RanGTP'-sample. If undetected in either 'input' or 'CRM1 w/o Ran', then the missing parameter was estimated as a lower bound (based on the detection sensitivity of our MS setup). Candidates detected only in 'CRM1+RanGTP' were not plotted (because for them, both parameters would have to be estimated). (C) Scatter plot is as in (B), but colour code is used to indicate the abundance in the 'CRM1+RanGTP'-bound sample.

We assume that most identified proteins from the categories A, B and 'low abundant' are direct CRM1 interactors that either carry a functional NES or occur in stable complexes with NES-containing proteins. Thus, we identified ≈1000 probable cargoes each in *Xenopus* and human as well as ≈700 in yeast. This is far more than identified for any other NTR (see e.g. *Kimura et al., 2013*), and represents ≈20% of all detectable nuclear or cytosolic proteins, only a small fraction of which (≈ 10%) had been proposed to be associated with CRM1 before (see *Supplementary file 5*). This suggests that CRM1 serves a far larger number of cargoes than previously assumed. To estimate our positive discovery rate of direct CRM1 cargos, we tested a subset of these candidates below.

Of course, nucleoporins, and FG Nups in particular, were also identified as CRM1 ligands. We consider them, however, as part of the transport machinery and not as cargoes. Some of them bind CRM1 very strongly, for example, the human or *Xenopus* Nup214•88•62 complex or Nup358 (see *Figure 3b*, and e.g. *Fornerod et al., 1997b*; *Engelsma et al., 2004*).

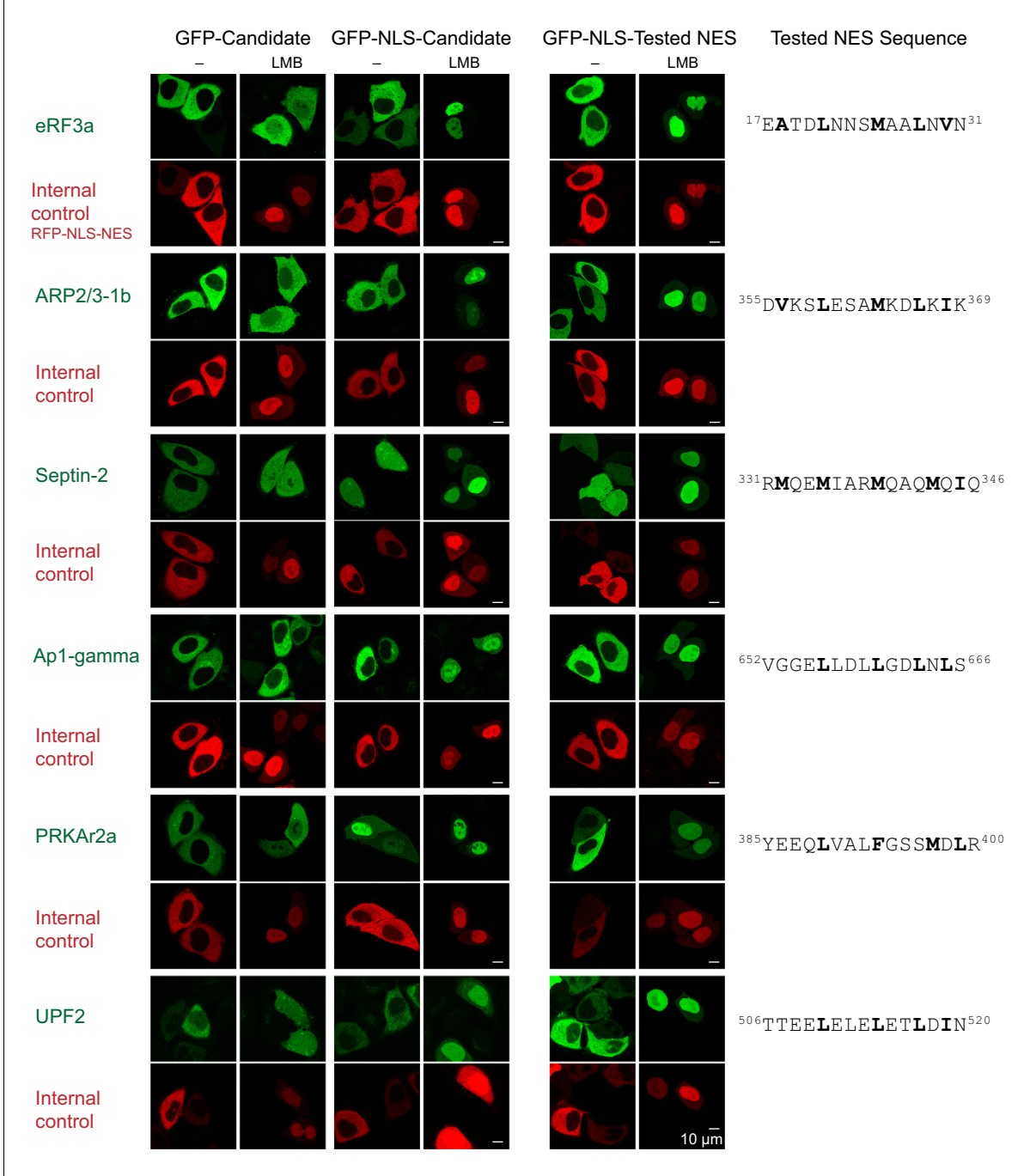

**Figure 4.** Validation of *Xenopus* CRM1-cargo candidates and identification of NESs. HeLa cells were transfected to express GFP- or GFP-NLS-fused candidate proteins, then incubated with or without the CRM1 inhibitor leptomycin B (LMB), fixed, and analysed by confocal laser scanning microscopy (CLSM). The co-transfected RFP-NLS-NES was detected in a separate channel as a control for the LMB-effect. Tested candidates: eukaryotic peptide chain release factor eRF3a (Q91855), subunit 1b of the ARP2/3 complex (Q6GNU1), Septin-2 (B7ZR20), Ap1-gamma subunit of the clathrin-associated adapter complex (Q6GPE1), the cAMP-dependent kinase type II-alpha regulatory subunit pRKAr2a (F7CZT8), and the regulator of nonsense transcripts UPF2 (Q498G1). UniProt entry names are given in parentheses. Figure also shows sequences of identified NESs, and their validations as transfected GFP-NLS-fusions with an LMB-sensitive cytoplasmic localisation.

On the other end of the distribution, we identified ≈700 proteins in *Xenopus*, ≈900 in human and ≈600 in yeast, which were strongly selected against in the 'CRM1+RanGTP'-bound fractions (***Supplementary files 2–4***, ***Figure 3***). These represent proteins, whose nucleocytoplasmic

partitioning is probably not directly affected by CRM1. Metabolic enzymes (of e.g. glycolysis, the pentose phosphate pathway etc.), protein folding factors, and exclusively nuclear proteins are over-represented in this 'non-binder' category.

In between, we found a broad zone of 'ambiguous' proteins, which actually represent a continuum. Some of them bound still very specifically to the export-form of CRM1 (according to the 'minus RanGTP'-control); yet, the binding was weak. We assume that these proteins become only transiently CRM1 cargoes (e.g. in response to the addition or removal of a post-translational modification) or that they only transiently associate with *bona fide* CRM1-cargoes. On the other end of the 'ambiguous' category, there are proteins, which appear to be 'CRM1 non-binders'. However, their low abundance in the starting extract precluded any reliable judgement of how strongly they were selected against in the 'CRM1+RanGTP'-bound fraction.

## Validation of identified CRM1 cargo candidates

The data set contains most of the previously well-validated CRM1 cargoes, such as the nuclear import adapter snurportin, or the nuclear export adapters NMD3 (*Ho et al., 2000*) and PHAX (*Kitao et al., 2008*). The vast majority of hits ($\geq$ 90%), however, were so far not linked to CRM1-mediated nuclear export. We therefore decided to verify a sample of candidates according to a common scheme.

The first was the *Xenopus* translation termination factor eRF3a. Its GFP-fusion was exclusively cytoplasmic in transfected HeLa cells (*Figure 4*), which is consistent with its experimentally determined localisation in *Xenopus* oocytes (*Supplementary file 1*). A CRM1-block by 10 nM leptomycin B, however, caused the fusion to equilibrate between nucleus and cytoplasm, suggesting a significant nuclear entry rate and rapid CRM1-dependent retrieval in undisturbed cells.

We also transfected a fusion that included an SV40 nuclear localisation signal (GFP-NLS-eRF3a) to enforce a faster nuclear entry, which made the fusion indeed exclusively nuclear following Leptomycin B-treatment. In undisturbed cells, however, we observed a still nearly exclusively cytoplasmic localisation, suggesting that the eRF3a NES confers a considerably faster export from nuclei than nuclear import mediated by the (very strong) SV40 NLS. We identified this NES within the N-terminal unstructured region of eRF3a and confirmed its nuclear export activity by transfection assays as well (*Figure 4*).

eRF3a binds also purified CRM1 in a RanGTP-dependent manner (*Figure 5*), suggesting that the interaction is direct and not bridged by another factor. Due to its efficient binding from *Xenopus* oocyte extract to CRM1, eRF3a was classified as a 'category A' cargo. Next, we also confirmed a far lower scoring 'category B' candidate, namely the 1b subunit of the Arp2/3 complex, as a directly-binding, *bona fide* CRM1 export cargo with an NES at its extreme C-terminus. In the light of the

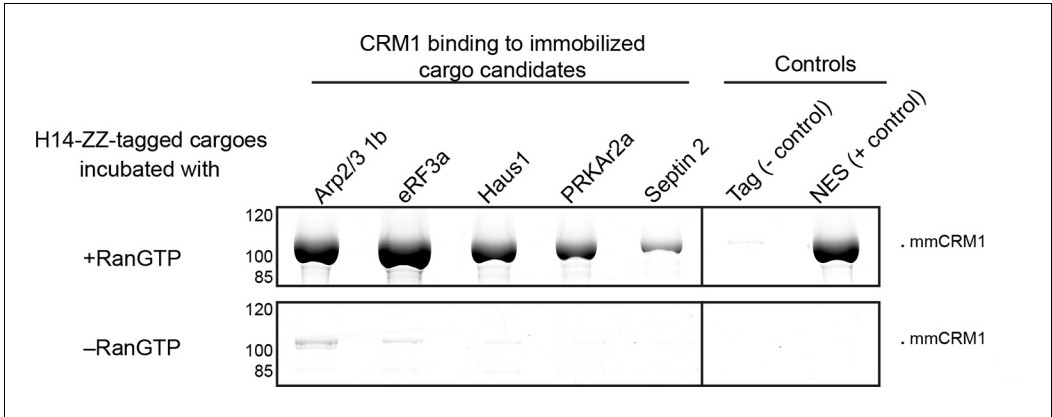

**Figure 5.** Identification of cargo candidates as direct CRM1-binders. The H14-ZZ-Sumo tagged candidate proteins ARP2/3 1b (Q6GNU1), eRF3a (Q91855), Haus1 (Q3B8L5), pRKAr2a (F7CZT8), Septin-2 (B7ZR20) were expressed in *E. coli*, purified, immobilised on anti-zz beads, and incubated with CRM1 in the absence or presence of RanGTP. Immobilised candidate proteins were released, after washing, by Sumo-protease cleavage and co-eluting materials were analysed by SDS-PAGE (Note that Septin-2 elution was less efficient than the others). An unfused H14-zz-Sumo module served as a negative and a fusion with a PKI-NES as a positive control for CRM1-binding.

rather weak CRM1-binding of Arp2/3-1b (*Supplementary file 2*), its NES turned out to be surprisingly strong. When fused to GFP, it behaved like a supraphysiological NES (*Engelsma et al., 2004*) and produced pronounced transport intermediates at NPCs (best visible in weakly expressing cells). This difference is, however, plausible in the context of the Arp2/3 complex structure (pdb 1K8K; *Robinson et al., 2001*), which shows this NES packing against the rest of the chain. The rather loose packing and high local B-factor suggest, however, that this NES region is sufficiently mobile to get transiently exposed for subsequent CRM1-binding.

In total, we tested 29 candidates from *Xenopus*, human and yeast, and validated 23 of them positive (*Figures 4*, *6*, *7* and *8*), suggesting that the majority of hits represent indeed CRM1 cargoes. Negative cases where, for example, the exclusively nuclear replication factor C (subunit 3) or xDDX6. Explanations could be an issue with NES-modulating post-translational modifications, a transport-independent interaction with CRM1 or that another subunit in a larger complex accounts for CRM1-binding.

We reasoned that the latter scenario might apply to xDDX6, which has been reported to interact with Lsm14 (isoforms a and b; *Tanaka et al., 2006*; *Arthur et al., 2009*). Indeed, the transfected Lsm14b-GFP fusion was exclusively cytoplasmic, but shifted upon leptomycin B treatment to a nuclear localisation. Given that Lsm14 had even better CRM1-binding scores than xDDX6 and that the two factors were recovered in ≈1:1 stoichiometry within the 'CRM1+RanGTP'-bound fraction (*Supplementary file 2*), it is most likely that Lsm14 is the direct CRM1-binder, while xDDX6 piggy-backs. Similar considerations will probably apply to numerous other cargo candidates that occur in complexes with other proteins. *Supplementary files 2–4* provide the information for interpreting such cases, because they list not only the already mentioned binding-scores, but also estimate the molar ratios in which cargo candidates were recovered in the 'CRM1+RanGTP'-bound fractions.

## Leakage of large 'CRM1 non-binders' into the nucleus

The sheer number of CRM1 cargoes already suggests a very broad impact of this nuclear export pathway on cellular physiology. Yet, there is a remarkable bias towards or against individual functional categories (*Figure 9*). Highly abundant metabolic enzymes (including glycolytic enzymes), for example, are greatly under-represented amongst the CRM1 cargoes. Here, it is remarkable that many of these CRM1 non-cargoes, including enzymes that are part of large complexes, show a rather even distribution between cytoplasm and nucleus of the *Xenopus* oocyte without having a detectable NLS (*Supplementary file 2*, sheets 'glycolysis' and 'metabolic enzymes'). This suggests that a large size alone cannot guarantee a cytoplasmic confinement.

## CRM1 and translation

Translation initiation factors represent the other extreme (*Supplementary files 2–4*, sheets 'Translation factors'; *Figure 9B*). eIF2, eIF2B, eIF4B, eIF4G, eIF5 and eIF5B all behaved like high-scoring cargoes in either *Xenopus*, human or yeast, which coincides nicely with their complete exclusion from the nuclei of *Xenopus* oocytes. We assume that CRM1 serves here the purpose of suppressing intra-nuclear translation and possibly also avoiding an interference with ribosome biogenesis if translation factors bound pre-maturely to ribosomal assembly intermediates.

Leakage from the cytoplasm into the nuclear compartment should be fast for small individual proteins, but slow for large entities. It is therefore remarkable that even very large translation factor complexes behaved like nuclear export substrates, examples being the 125 kDa eIF2αβγ complex (150 kDa with tRNA) or the 270 kDa eIF2Bαβγδε complex (*Supplementary files 2–4*, sheets 'Translation factors'). This suggests that even the presumably slow leakage of large cytoplasmic assemblies into nuclei can be so deleterious that countermeasures are required.

None of the translation elongation factors appeared to be a convincing CRM1 cargo, possibly because appending an NES to the elongation factors might be incompatible with efficient translation. Nevertheless, they are also subject to exportin-mediated nuclear export. Human EF1A was previously identified as the most prominent export cargo of Xpo5, while the elongation factor for proline-rich regions, eIF5A, is exported by Xpo4 (*Lipowsky et al., 2000*; *Bohnsack et al., 2002*; *Calado et al., 2002*). It is thus well possible that other translation factors that fail to interact with CRM1 also use a more specialized nuclear export pathway.

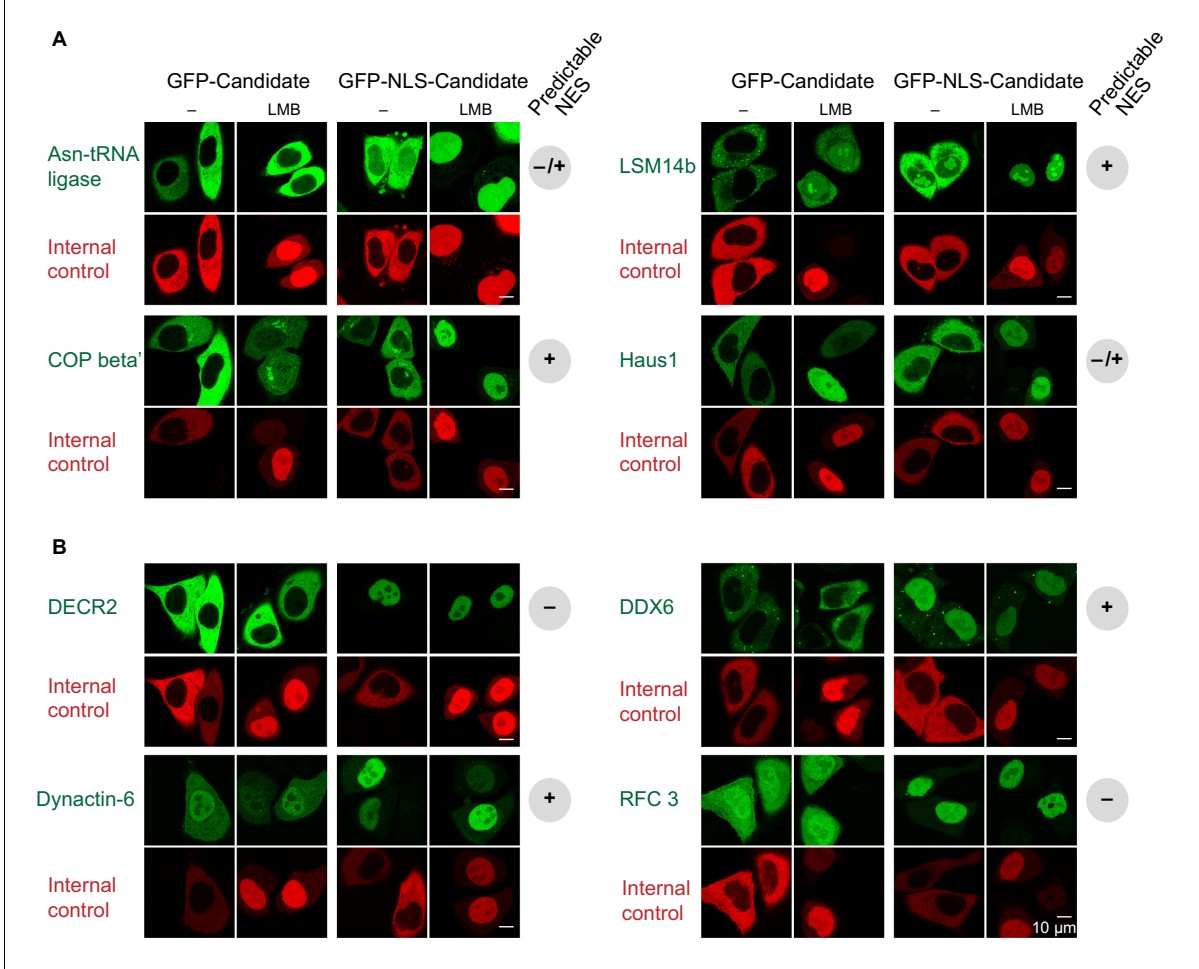

**Figure 6.** Validations of additional CRM1 cargo candidates from *Xenopus*. Analysis was as in *Figure 4*. (**A**) Tested candidates that behave like true CRM1 cargoes: Asn-tRNA ligase (Q6DD18), LSM14b (L14BB), COP beta' (Q7ZTR0), and Haus1 (Q3B8L5). (**B**) Tested candidates that are not CRM1 cargoes: the peroxisomal 2,4-dienoyl-CoA reductase DECR2 (Q6GR01), the RNA helicase DDX6, dynactin 6 (Q6IRC3) and the replication factor complex subunit RFC 3 (Q4QQP4). DDX6 had been in cargo category A, but probably requires Lsm14 (see panel A and main text) for CRM1 interaction. We assume an analogous scenario for DECR2. Dynactin 6 is in category 'ambiguous' and was therefore not considered a CRM1 cargo in the first place.

The translation termination factor eRF3a is again a *bona fide* CRM1 cargo (see *Figure 4*), as is eRF1 (*Supplementary file 2* and *Bohnsack et al., 2002*). Furthermore, CRM1-dependent nuclear depletion also applies to several translation-associated factors, such as the signal recognition particle SRP, where SRP54 appears to be the NES-carrying component (*Figure 7*), or the human start-methionine aminopeptidase MetAP2, which binds translating ribosomes and cleaves the start methionine from nascent polypeptides. The normally strictly cytoplasmic MetAP2 accumulates upon leptomycin B treatment in nucleoli (*Figure 7*), not only validating it as a true CRM1 cargo, but also suggesting that it can bind to assembling ribosomal subunits prematurely and possibly interfere with the maturation process.

## CRM1 and ribosome biogenesis

CRM1 is known to be essential for ribosome biogenesis. It exports pre-60S ribosomal subunits (rSUs) through the export adapter Nmd3 (*Ho et al., 2000*), while Ltv1 and Rio2 behave like export adapters for yeast 40S rSUs (*Seiser et al., 2006*; *Fischer et al., 2015*). Yeast Arx1 and the Mex67•Mtr2 dimer also bind pre-60S rSU subunits and promote export through direct interactions with FG repeats (*Bradatsch et al., 2007*; *Yao et al., 2007*). Furthermore, Ecm1, Bud20, Alb1, Tif6, Rlp24, Nog1, and Mtr4 are known to escort 60S rSUs, while Nob1 and Enp1 have been shown to accompany 40S rSUs to the cytoplasm, without remaining a constituent of mature ribosomes (reviewed in

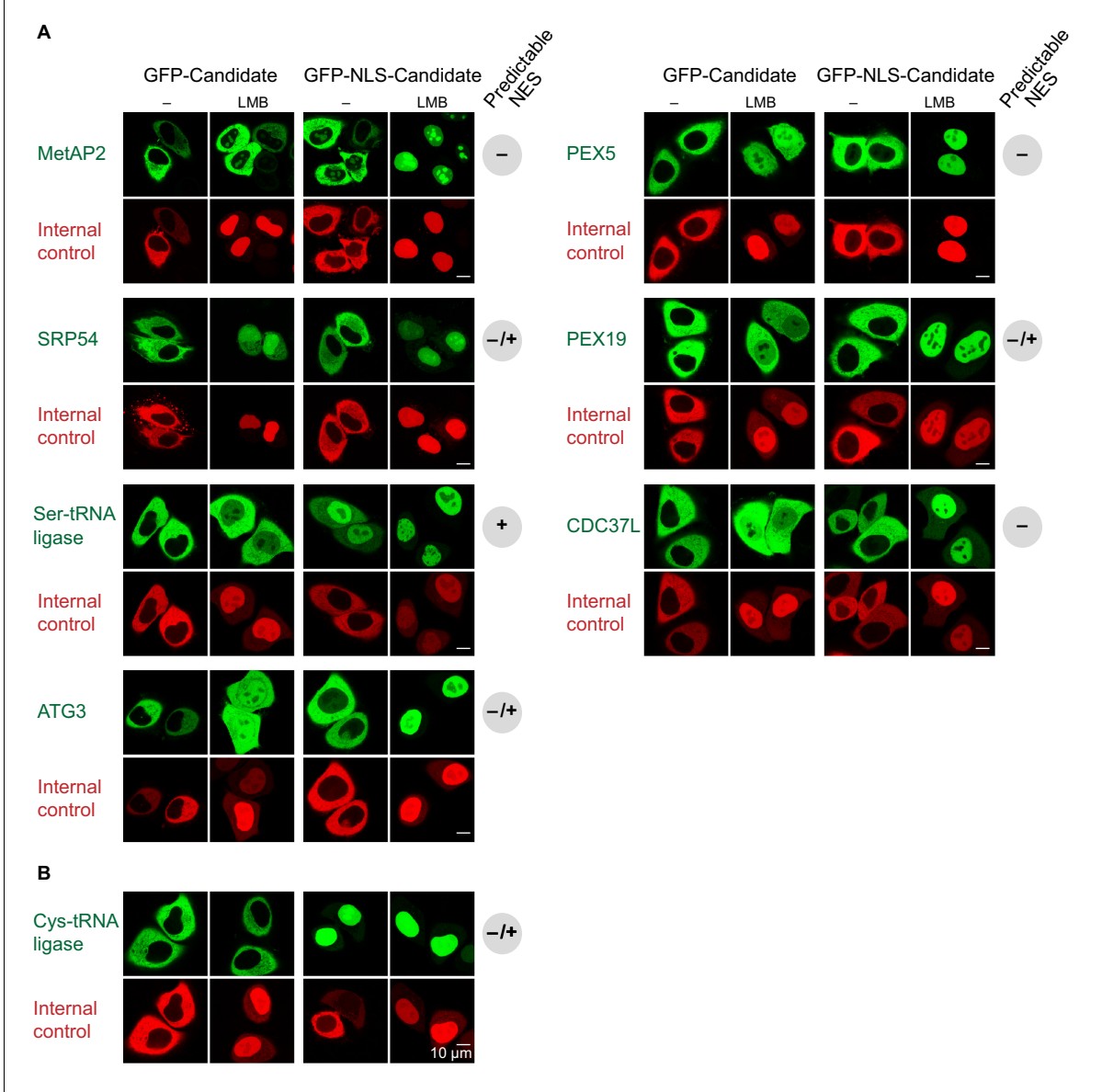

**Figure 7.** Validation of human CRM1 cargo candidates. Analysis was as in *Figure 4*. UniProt identifiers correspond either to abbreviated protein names or are given in parentheses. (**A**) Positively tested CRM1 cargoes: the co-translational methionine aminopeptidase MetAP2 (MAP2), PEX5, SRP54, PEX19, the Ser-tRNA ligase (SYSC), CDC37L (CD37L), and ATG3. (**B**) The Cys-tRNA ligase (SYCC) was classified as a 'CRM1-non-binder' and accordingly shows a CRM1-independent nuclear exclusion.

The following figure supplement is available for figure 7:

**Figure supplement 1.** Validation of human CRM1 non-binders.

*Panse and Johnson, 2010*; *Thomson et al., 2013*). We now found that yeast Enp1 (essential nuclear protein 1) behaves like an autonomous CRM1 substrate (*Figure 8*), suggesting that it might actually act as yet another adapter for CRM1-mediated export of 40S rSUs. The use of multiple export adapters, as in here, might not just be an issue of robustness and redundancies, but a more fundamental requirement for making such large cargoes sufficiently 'soluble' in the gel-like permeability barrier of NPCs (*Ribbeck and Görlich, 2002*; *Schmidt and Görlich, 2015*).

Furthermore, our analysis revealed that additional predominantly nuclear, ribosomal biogenesis factors behave like CRM1 cargoes, namely: Brx1, Pno1, Tsr1, Rpf1, Rlp7, Ytm1, Tsr3, Rix7, Erb1,

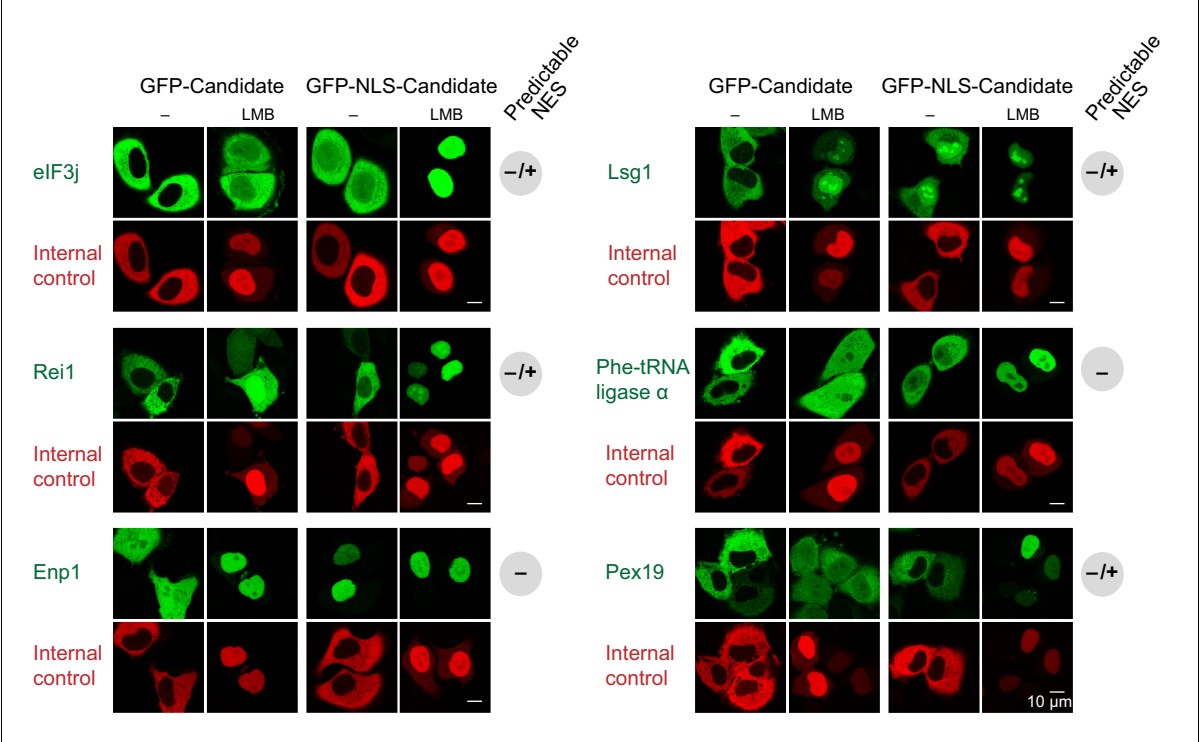

**Figure 8.** Validation of CRM1 cargo-candidates from *S. cerevisiae.* Analysis was as in *Figure 4* and included: the translation initiation factor eIF3j, the ribosome biogenesis factors LSG1, REI1, and ENP1, the α-subunit of the Phe-tRNA ligase (SYFA), as well the peroxisome biogenesis factor PEX19. This positive validation of yeast cargoes in HeLa cells also emphasises the extreme conservation of NES-recognition by CRM1.

Nsa1, Ssf1, Dim1, Nol10, Loc1, and Rpf2 (see *Supplementary file 4* sheet 'Ribosome biogenesis'). Thus, they might also escort rSUs to the cytoplasm, and possibly facilitate the export process by providing additional binding sites for CRM1.

Newly exported 40S and 60S rSUs acquire translation competence only after a series of maturation steps in the cytoplasm. In the case of yeast 60S rSUs, this involves the activities of Drg1/ Afg2, Rei1, Reh1, Jjj1, Yvh1, Lsg1, Efl1/ Ria1 as well as of Sdo1 (reviewed in *Panse and Johnson, 2010*; *Thomson et al., 2013*). While some of these might get loaded already inside the nucleus, it appears that Drg1, Rei1, Reh1, Lsg1, and Ria1 act exclusively in the cytoplasm. All these factors showed a strong and strictly RanGTP-dependent interaction with CRM1 (*Supplementary file 4* sheet 'Ribosome biogenesis'). This suggests that CRM1 keeps them cytoplasmic, possibly to avoid the occurrence of translation-active ribosomes inside the nucleus (at least Lsg1 and Rei1 behave like true CRM1 cargoes, see *Figure 8*). A cytoplasmic confinement of these factors appears crucial for yet another reason: They displace the export mediators Arx1, Nmd3, Mex67, and Mtr2 from newly exported 60S species (*Loibl et al., 2014*). A premature intra-nuclear displacement by mis-localized cytoplasmic maturation factors could thus abort ribosomal export.

Cytoplasmic maturation also involves the incorporation of additional ribosomal proteins, a prominent example being rpS3. rpS3 is highly abundant in the 'CRM1+RanGTP'-bound fraction, and thus perhaps hindered (directly or indirectly) by CRM1 to bind 40S rSUs prematurely already inside nuclei. The same might apply to rpL24a.

### Aminoacyl tRNA ligases

These enzymes are essential for translation, and a cytoplasmic confinement of tRNA-charging would also contribute to the exclusion of intranuclear translation. Nevertheless, nuclear aminoacylation has been proposed as a proof-reading step for correct pre-tRNA processing and maturation prior to export (*Lund and Dahlberg, 1998*; *Sarkar et al., 1999*). This proposal was based on the observations that blocking aminoacylation in yeast results in nuclear accumulation of tRNA and that a similar

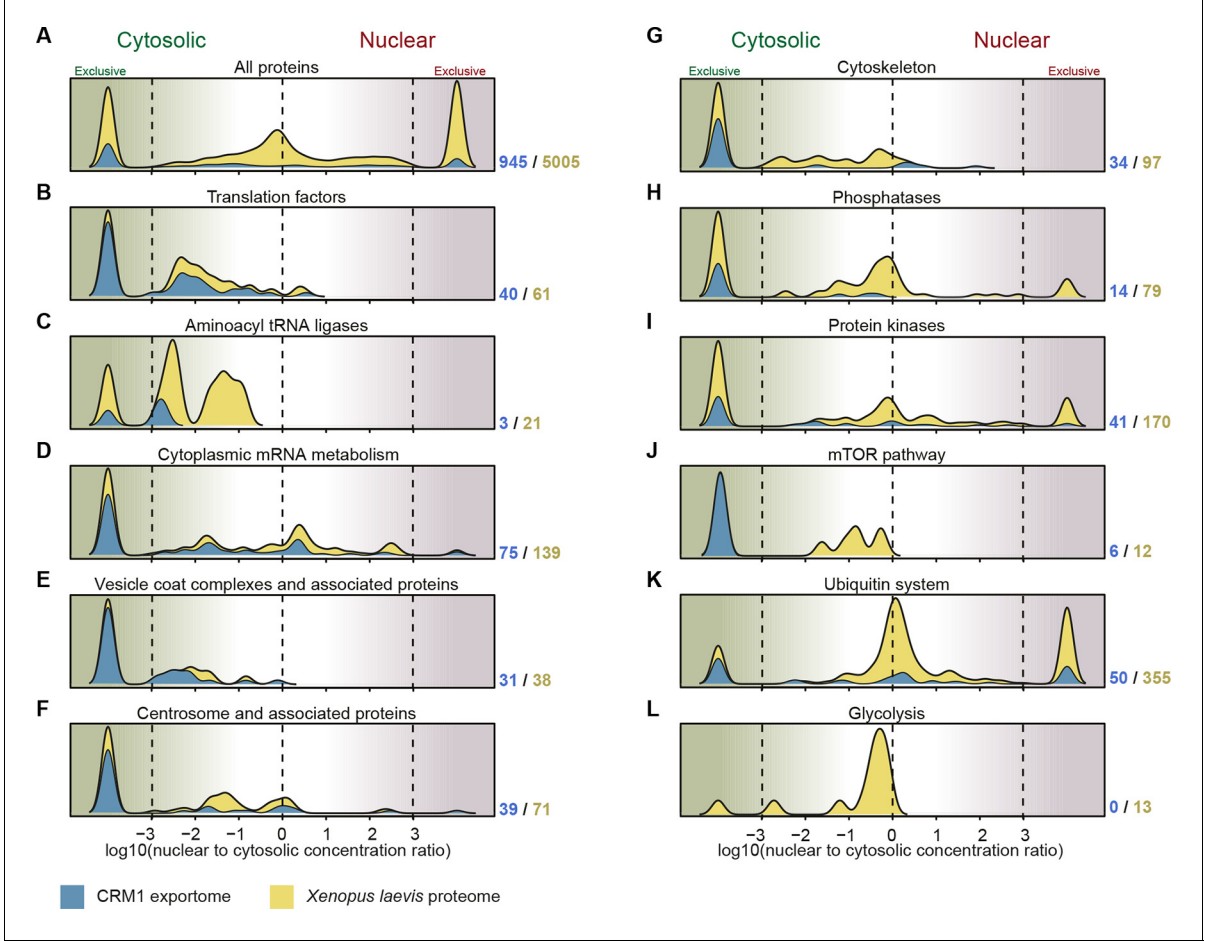

**Figure 9.** Correlation between functional groups, nucleocytoplasmic partitioning and CRM1-interaction. (A) Panel shows a 'density plot' to illustrate how many unique proteins (see *Figure 1D*) show a given N:C partition coefficient in *Xenopus* oocytes. The yellow area covers all proteins, the blue area only proteins that belong to CRM1 cargo categories 'A'-'C'. (B–L) Density plots are analogous to (A), but each panel represents just one functional group and the curves were re-scaled to account for the smaller number of proteins in a given group. Functional groups were initially defined by KEGG BRITE hierarchies and then manually refined (see *Supplementary file 2* for included proteins).

treatment in *Xenopus* oocytes prevents tRNAs, which had been injected into the nucleus, from reaching an exclusively cytoplasmic steady state distribution. There are, however, also arguments against this scenario, foremost that a loss of cytoplasmic retention of tRNA by translation elongation factor eEF1A (which binds only aminoacylated tRNA) would explain the phenotypes as well. Furthermore, a structural and functional analysis of tRNA•Xpo-t•Ran export complexes revealed that this exportin proof-reads a correct 3'-CCA end (*Arts et al., 1998*; *Lipowsky et al., 1999*), but cannot sense aminoacylation (*Cook et al., 2009*). The latter applies also to the alternative tRNA exporter Xpo5 (*Bohnsack et al., 2002*; *Calado et al., 2002*).

To clarify this issue at least for *Xenopus* oocytes, we analysed the nucleocytoplasmic distribution of the aminoacyl tRNA ligases and found a strong cytoplasmic bias (*Figure 9C*; *Supplementary file 2*, sheet 'Aminoacyl tRNA ligases'). The cytoplasmic concentration exceeded the nuclear one by more than a factor of 100 in most cases, and even the least excluded ones had ≈10 times higher levels in the cytoplasm than in the nucleus. This makes is rather unlikely that nuclear aminoacylation is a general proof-reading criterion prior to tRNA export. At least the Ser, Thre, and Asn tRNA-ligases showed a strong RanGTP-dependent interaction with CRM1. We tested the *Xenopus* Asn tRNA ligase by transfection assays and observed a perfect CRM1-dependent nuclear exclusion (*Figure 6*). Human Ser tRNA ligase and the α-subunit of the yeast Phe tRNA ligase behave the same way (*Figures 7* and *8*), suggesting that cells make a true effort to keep tRNA aminoacylation cytoplasmic.

We also tested the human Cys tRNA ligase that failed to interact with CRM1 (*Supplementary file 3*). Nevertheless, the transfected GFP-fusion protein showed a perfect nuclear exclusion, which was also not impaired by leptomycin B-treatment (*Figure 7B*). How this cytoplasmic confinement is maintained is still unclear, but the formation of larger complexes or alternative export pathways are plausible possibilities. In contrast to a CRM1-dependent confinement, the cytoplasmic localisation can, however, not be maintained against a fused SV40-type NLS. These transfection experiments also exemplify the validation of a 'CRM1-non-binder' (for more such validations, see below).

## CRM1 and mRNA degradation

We identified several high-scoring CRM1 cargoes amongst cytoplasmic mRNA degradation factors, components of P-bodies and stress granules. These include Upf1, Upf2 and Upf3 (*Supplementary files 2–4* and *Figure 5*), which function in nonsense-mediated decay (NMD) of incorrectly spliced mRNA during a pioneering round of translation (reviewed by *Popp and Maquat, 2013*), as well as the de-capping enzymes (DCP1 and DCP2) and enhancers of de-capping (EDC proteins), which typically reside within P-granules and initiate mRNA degradation (reviewed by *Parker and Sheth, 2007*). Thus, CRM1 probably enables cells to control nuclear and cytoplasmic RNA turnover independently from each other. This CRM1 function appears very well conserved from yeast to human.

## CRM1, vesicular transport and autophagy

So far, no connection has been made between CRM-mediated nuclear export and vesicle formation along the secretory pathway (reviewed by *Kirchhausen, 2000*). Yet, we observed that COPI and COPII coat proteins as well as, e.g., the AP-1 and AP2 adapter complexes or the AP180 clathrin coat assembly protein behave like CRM1 cargoes (*Supplementary files 2–3*, sheets 'Vesicle coat proteins'; as well as *Figures 4* and *6*). They are also extremely well excluded from the oocyte nucleus (*Figure 9E*). A formation of intranuclear vesicles was so far observed only in oocytes of rather exotic species, such as the ascidian *Botryllus schlosseri* (*Manni et al., 1994*). The absence in other cell types can now be explained by an active depletion of the vesicular budding machineries from the nuclear interior.

The data also suggest an unanticipated connection of CRM1 to autophagy (reviewed in *Reggiori and Klionsky, 2013*). Atg1, Atg13, Atg17, and Vps30 are all required for autophagy in yeast and all of them are high-scoring CRM1 cargoes (*Supplementary file 4*, sheet 'Autophagy'). It thus appears as if CRM1 counteracted an initiation of autophagy from the nuclear interior. The situation is very similar in human cells, though the spectrum of CRM1-interacting autophagy components is slightly different (*Supplementary file 3* sheet 'Autophagy'). For example, here the ATG8-conjugating enzyme ATG3 is a major CRM1-cargo (*Figure 7*).

## CRM1 and post-translational transport to peroxisomes

Our data set also revealed high-scoring CRM1 cargoes that make an unexpected link between nuclear export and protein import into peroxisomes (reviewed in *Ma et al., 2011*): The peroxisomal targeting (PTS1) receptor Pex5 was not only identified as a high-scoring CRM1-binder, but also showed a strictly CRM1-dependent nuclear exclusion (*Figure 7*). This connection might point to a general challenge for post-translational transport from the cytosol, namely that diffusive transport will not necessarily lead to the destination organelle, but also to and possibly into nuclei. In the case of peroxisomal proteins, this poses a particular danger as many of them produce reactive oxygen species that might damage the genome. A first line of defence against such incidents is a trapping of peroxisomal proteins by dedicated targeting receptors. The resulting receptor•substrate complexes, however, still need to reach peroxisomes by diffusive transport. If this fails and the complex ends up inside nuclei, then CRM1-mediated export can rectify the problem and give the targeting complex another chance to reach its correct destination.

PEX19 from yeast or human is another example, which behaves like a perfect CRM1 cargo and shows a CRM1-dependent nuclear exclusion (*Figures 7* and *8*). PEX19 targets membrane proteins to peroxisomes and allows pre-peroxisomes to bud from the ER (reviewed in *Ma et al., 2011*). A mistargeting of PEX19 to nuclei by a fused NLS causes interesting consequences, namely nuclear

accumulation of newly synthesised peroxisomal membrane proteins (*Sacksteder et al., 2000*). It now seems very likely that a loss of CRM1-mediated export would have the same effect.

## CRM1, cytoskeleton and centrosomes

Apart from a cytoplasmic confinement of e.g. VASP or the Arp2/3 complex (*Supplementary file 3*, *Figure 5*), it appears that CRM1 has only little direct impact on the actin cytoskeleton, which is consistent with the fact that metazoans possess a dedicated exportin (Xpo6) to deplete actin from their nuclei (*Stüven et al., 2003*).

*Xenopus* and human septins, in contrast, are highly abundant amongst the CRM1-bound proteins. We validated Septin 2 and found that leptomycin B treatment of transfected cells caused indeed a shift from a cytoplasmic to a nuclear localisation (*Figure 5*). Septins interconnect actin and microtubule networks, function in cytokinesis, formation of cilia and defence against pathogens (*Mostowy and Cossart, 2012*), and it appears that their nuclear accumulation needs to be actively suppressed.

The data also suggests some contribution of CRM1 to nuclear exclusion of α- and β-tubulin in metazoan cells (*Supplementary files 2* and *3*). We further found high-scoring CRM1 cargoes amongst the components of the centrosomes (*Figure 9F*), which function as microtubule organizing centres (reviewed by *Bornens, 2012*). These strong CRM1 interactors include the HAUS augmin complex, γ-tubulin and γ-tubulin complex components as well as a number of additional centriolar proteins (e.g. CEP41, CEP55, and CEP170). This could point not only to an active suppression of microtubule nucleation in metazoan interphase nuclei, but also to additional mitotic function of CRM1 (see *Arnaoutov et al., 2005*)—namely at centrosomes.

## CRM1 and regulatory proteins

Almost every aspect of cellular physiology is under the control of protein kinases, phosphatases or the ubiquitin/ proteasome system. Our data now suggest that CRM1 is more heavily involved in such regulatory circuits than previously thought. Alone in HeLa cells, we found ≈ 70 kinases as high-scoring CRM1 cargoes (*Supplementary file 3*), only a fraction of which had been described as such before. The new cargoes include heavily studied kinases such as protein kinase A, the interleukin-1 receptor-associated kinase 1, the pro-apoptotic serine/ threonine kinase 3, the tank-binding kinase 1, Raf-1, and several isoforms of casein kinase I and II. In addition, we identified numerous kinase regulators as new CRM1 cargoes, examples being the already mentioned regulatory subunit of PKA (PRKAr2a; *Figures 4* and *5*) or the TSC1•TSC2 complex (*Supplementary file 2*), which is a key component of the mTOR signalling pathway (*Laplante and Sabatini, 2009*). Such action of CRM1 will contribute to compartment-specific phosphorylation patterns, but in many cases also exert control by granting or denying those kinases access to their substrates.

Likewise, numerous phosphatases and components of the ubiquitin or the ubiquitin-like modifier system showed up as potential new CRM1 cargoes, alone in human cells 22 and 57, respectively (*Supplementary file 3*, *Figure 9H and 9K*).

## Identification of new NESs and impact on NES prediction

CRM1-dependent nuclear export signals are usually short (9–15 residues long) peptides with 4–5 hydrophobic Φ residues that are spaced according to characteristic patterns (*Wen et al., 1994*; *Fischer et al., 1995*; *Güttler et al., 2010*; *Xu et al., 2012a*). The most common one is a PKI-type NES. Several other spacings are also allowed, which can be explained either by such NESs binding in a different conformation to CRM1 (snurportin-type or Rev-type NES) or by skipping one Φ-residue in a 5Φ NES. A functional NES also needs to be solvent-exposed and not buried in a globular fold.

We applied these criteria to identify NESs in a set of validated, new CRM1 cargoes. We were successful in six cases (*Figure 5*), but failed with some others (e.g. Pex5, rpS3, Haus1, Cop beta', MetAP2, CDC37L, Enp1, Phe tRNA ligase α). These negative cases illustrate how difficult a reliable NES prediction still is. These difficulties originate from at least two problems. First, many published NESs turned out to be incorrect (see e.g. *Xu et al., 2012b* and discussion therein). NES prediction will therefore remain unreliable and produce frequent false-positive hits as long as it is based on an unreliable list of positive cases. Second, we probably miss true NESs, because we do not yet know all NES patterns.

Our study should now provide solutions to these limitations. First, we provide large test sets for benchmarking current and future NES predictions, first of all a total of ≈1300 'category A cargoes', which should contain an NES (in the case of oligomers at least one per complex). The subset of such true CRM1 cargoes, which lack a so far recognizable export signal, provides an ideal starting point for identifying NESs that conform to new patterns. We also found ≈2200 clear non-CRM1-binders that can serve as a negative control group for NES prediction.

The just mentioned non-binders include several cases (11) that have been listed as CRM1-cargoes in the NESdb (*Supplementary file 5*; *Xu et al., 2012b*). This could now point to the false-positives in the NESdb or to contaminating false-negatives in our dataset. To address this issue, we analysed seven of these conflicting cases and tested the behaviour of the corresponding GFP fusions in the HeLa cell transfection assay (*Figure 7—figure supplement 1*). All of them showed a considerable nuclear signal already in untreated cells and this nuclear signal did not further increase upon blocking CRM1 by leptomycin B. These seven re-tested candidates thus behaved as predicted from their classification as CRM1 non-binders, at least for the tested protein isoforms (cloned from HeLa cell cDNA), for the tested cell type (human HeLa cells) and under standard cell culture conditions.

An interesting twist to CRM1-mediated export is that it can be regulated by post-translational modifications. Cyclin B, for example, is kept cytoplasmic by CRM1 until prophase, when the NES gets inactivated by phosphorylation of adjacent serine residues and the protein suddenly accumulates inside nuclei (*Yang et al., 1998*). PHAX, the export adapter for U snRNAs, on the other hand, requires phosphorylation for an efficient interaction with CRM1 (*Kitao et al., 2008*). Given these examples, it will now be very interesting to obtain a global view on how phosphorylation impacts individual cargo-CRM1 interactions, for example, by testing in how far CRM1 selects for or against the corresponding phosphoforms.

## What necessitates an active sorting of obligatory cytoplasmic proteins?

Surprising outcomes of our study not only were the sheer number of CRM1 cargoes (≈ 1/4 of all detectable cytoplasmic and nuclear proteins), but also that the majority of cargoes are actually exclusively cytoplasmic proteins (*Figure 9A*; *Supplementary files 2–4*). This poses the questions of why there is a need to actively maintain a cytoplasmic localization and why evolution has not come up with a better barrier system?

The perhaps best answer is that is impossible to preclude nuclear entry of unwanted proteins in the first place, because there are several leakage routes into the nucleus. First, the NPC permeability barrier is *per se* probably imperfect and allows leakage: It is based on the sieving effect of reversibly crosslinked FG repeat domains and probably represents compromise in the sense that a tighter barrier would also restrict facilitated transport (see *Schmidt and Görlich, 2015* and discussions therein).

In addition, there are situations, where even the most perfect NPC barrier gets bypassed. The open mitosis in metazoans, for example, leads to a complete mixing of nuclear and cytoplasmic contents and requires an unmixing following reformation of the nuclear envelope. Nuclear proteins require re-import, while obligatory cytoplasmic proteins, such as translation factors, components of the vesicular transport machinery and many others, are obviously subject to active nuclear export. Another source of leakage might be NPC assembly, where inner and outer nuclear membrane might already fuse to a pore, before all FG Nups are in place to maintain a permeability barrier.

Finally, cells should survive a temporary damage of their nuclear envelope, which is known to occur rather frequently in certain cultured cancers cells (*Hatch et al., 2013*), but might also happen in normal cells, in particular when exposed to mechanical stress. Such incidents not only require a repair of the NE, but also a rapid unmixing of nuclear and cytoplasmic contents by active import as well as by active export. Thus, the barriers of the NE alone cannot maintain the compartment identities. Instead, a robust separation of nuclear and cytoplasmic contents requires active corrective mechanisms, whereby the exportin CRM1 appears to play a very central role.

## Materials and methods

### Microdissection of *Xenopus laevis* oocytes

Manual microdissection of the oocytes was performed in '5:1/HEPES buffer' (10 mM HEPES/ KOH pH 7.5, 83 mM KCl, 17 mM NaCl, supplemented with Roche complete protease inhibitor (EDTA-

free) as previously described (*Liu and Liu, 2006*). Isolated nuclei were gently washed several times, and proteins were recovered by ethanol precipitation.

The respective cytoplasmic fractions were diluted with '5:1/HEPES buffer' and homogenized with a pestle. Pigments, yolk and membranes were removed from the extract by two rounds of centrifugation (17,000 g, 15 min, 4°C). Note that thereby also insoluble protein complexes, such as intermediate filaments were removed, and that our analysis explicitly aimed at the nucleocytoplasmic partitioning of soluble complexes and proteins. We therefore also excluded proteins that co-purified with nuclei through their association with the nuclear envelope (NE). Such proteins were identified and 'flagged' by a manual dissection of nuclei into nuclear interiors and crude NEs (kindly performed by Volker Cordes) and by asking which 'nuclear' proteins did not fractionate with the nuclear interiors.

A total of 775 unique proteins (see below) were flagged as possibly not being cytosolic or intranuclear, 748 of them were removed from the list of *Supplementary file 1*, while 27 were kept in the final list because they were manually qualified as cytosolic or intranuclear proteins according to previous literature, or according to their subcellular localization at Human Proteome Atlas (*Uhlén et al., 2015*).

Quantitative mass spectrometric analysis of the obtained nuclear and cytosolic fractions involved three biological replicates with two technical replicates each. For quantification of nucleocytoplasmic partitioning, we considered that a nucleus is 10-fold smaller in volume than a yolk-free cytoplasm. Therefore, we compared for each analysis ≈60 nuclei with ≈6 cytoplasms.

## CRM1 affinity chromatography

In these experiments, we used biotinylated CRM1 versions carrying an Avi-tag, which is an optimised biotin-acceptor sequence for enzymatic biotinylation by BirA (*Schatz, 1993*).

HeLa S100 extract (*Abmayr et al., 2006*) and *Saccharomyces cerevisiae* whole cell extract (*Gottschalk et al., 1999*) were kindly supplied by the group of Reinhard Lührmann. *Xenopus laevis* oocyte extract was prepared as described (*Leno et al., 1996*).

*Xenopus laevis* oocyte, *Saccharomyces cerevisiae* and cytosolic HeLa extracts were diluted 1:5 in binding buffer (20 mM HEPES/ NaOH pH 7.5, 90 mM KAc, 2 mM MgOAc, 250 mM Sucrose, 5 mM DTT) and cleared by a 1 hr centrifugation step in a S55A rotor at 4°C and 37,000 rpm. Supernatants were incubated with Phenyl-Sepharose (low substitution) for the selective depletion of endogenous nuclear transport receptors as described previously (*Ribbeck and Görlich, 2002*). The flow-throughs were incubated with 0.1 µg/ml RNAse A for 20 min on ice (to detach 'indirect' cargoes that interact through RNA with direct ones) and then 100 units/ml RNasin (Promega) were added.

1 ml of a such treated extract was supplemented with 5 µM RanGTP (hsRanQ69L$^{5-180}$), centrifuged in a S45A rotor for 1 hr at 4°C at 37,000 rpm. 500 pmoles mmCRM1 or scCRM1 were immobilized on 20 µl streptavidin agarose beads (Sigma Aldrich). Free biotin-binding sites were quenched thereafter with 1 mM biotin, and the beads were rotated for 3 hr at 4°C with the RanGTP-supplemented extract. Beads were then washed three times with 500 µl binding buffer. Bound material was eluted with 60 µl SDS sample buffer at 45°C. Minus CRM1 and minus RanGTP controls were processed analogously. Input extracts as well as elutions were analysed by SDS-PAGE followed by Coomassie-staining and/or analysed by mass spectrometry as described below.

## Sample preparation for mass spectrometry

An overview of the sample preparation for mass spectrometry and LC-MS/MS instrumentation:

| | Analysis of CRM1 binders (all species) | Analysis of nucleocytoplasmic partitioning | | |
| --- | --- | --- | --- | --- |
| | | Replicate 1 | Replicate 2 | Replicate 3 |
| Sample preparation | SDS-PAGE, digestion with trypsin | SDS-PAGE, digestion with trypsin | SDS-PAGE, digestion with trypsin | Digestion with Lys-C and trypsin, reverse phase HPLC at pH 10 |
| LC-MS instrumentation | Dionex Ultimate 3000 HPLC Q-Exactive HF | EASY nLC-1000 Q-Exactive | Dionex Ultimate 3000 HPLC Q-Exactive HF | Dionex Ultimate 3000 HPLC Orbitrap Fusion |

To estimate absolute protein concentrations, Universal Proteomics Standard-2 (UPS2; Sigma-Aldrich) was added to the analysed samples (at a 1:10 (w/w) ratio between the standards and the total sample protein). Since UPS2 is based on human proteins, this standard was only employed for the non-human samples.

In one workflow, proteins were first separated by SDS-PAGE (4-12% Bis/Tris gradient mini-gel, NuPAGE, Novex) and visualized by colloidal Coomassie-staining. Proteins were then in-gel digested as described before (*Shevchenko et al., 2006*) with minor modifications. Briefly, proteins were reduced with 10 mM DTT for 30 min at 55°C, and then alkylated with 55 mM iodoacetamide (IAA) in 50 mM ammonium bicarbonate (BC) for 20 min at 26°C in the dark. Protein digestion was performed overnight at 37°C at a 1:50 (w/w) trypsin (Promega #V5111) to protein ratio. Following digestion, peptides were extracted from the gel pieces, and concentrated by vacuum evaporation of the solvent in a SpeedVac to near dryness. Dried peptides were dissolved in 20 µl of 1% (v/v) formic acid, and 6 µl were analysed LC-MS/MS for each technical replicate.

In the second workflow, ethanol-precipitated proteins were dissolved in 1% (v/v) RapiGest SF surfactant (Waters # 186002122) at 70°C for 10 min. Proteins were then reduced with 5 mM DTT in 50 mM BC for 30 min at 50°C, and alkylated with 10 mM IAA in 50 mM BC for 20 min. Excess IAA was reacted with an additional 5 mM DTT at RT for 20 min. Proteins were first digested with Lys-C (Roche, 1:100 enzyme to protein ratio) for 4 hr at 37°C, then overnight with trypsin at a final RapiGest concentration to 0.1%. Following digestion, the samples were acidified with trifluoroacetic acid (final concentration of 1%, v/v, 37°C, 1 h) to break down the RapiGest surfactant, the resulting by-products were pelleted by centrifugation (13,000 rpm, 15 min, RT), and the supernatant containing the digested peptides was transferred to a new tube. Peptides were desalted on reversed phase-C18 solid-phase extraction cartridges (SPE; SepPak, Waters) and concentrated in a SpeedVac to near dryness. Then, peptides were resuspended in 10 mM ammonium hydroxide (pH 10) and loaded onto a reverse phase HPLC column (XBridge C18, Waters, 3.5 µm, 1.0 mm x 150 mm) and eluted in a 5-35% (v/v) acetonitrile gradient at a flow rate of 60 µl/min. 45 initial fractions were collected, which were combined into 17 peptide pools. Each pool was concentrated as described above and dissolved in 20 µl 1% FA (v/v). 6 µl each were then analysed by LC-MS/MS for a technical replicate.

## LC-MS/MS Q-Exactive analysis

First, extracted peptides were loaded onto an in-house packed C18 'trapping' column (0.15 mm x 20 mm, Reprosil-Pur 120 C18-AQ 5 µm, Dr. Maisch GmbH, Germany). Then a second C18 column was connected in tandem (an analytical C18 capillary column; 0.075 mm x 250 mm column self-packed with 3 µm Reprosil-Pur 120 C18-AQ). Peptides were then eluted using a 105 min linear gradient (5– 35% acetonitrile in 0.1% FA at 300 nl/min) on an EASY nLC-1000 system in-line coupled to a Q Exactive hybrid quadrupole/orbitrap mass spectrometer (Thermo Scientific, Dreieich). The instrument was operated in data-dependent acquisition mode with a survey scan resolution of 70,000 at m/z 200 and an AGC target value of $1 \times 10^6$. Up to 15 of the most intense precursor ions with charge state 2 or higher were sequentially isolated at an isolation width of 2.0 m/z for higher collision dissociation (HCD) with a normalized collision energy of 25%. Dynamic exclusion was set to 30 s to avoid a repeating sequencing of the same precursor ion.

## Q-Exactive HF and Orbitrap Fusion analysis

The LC setup was as described above, but a Dionex Ultimate 3000 HPLC (Thermo Scientific, Dreieich) and a 350 mm capillary C18 column were used.

Both mass spectrometers were operated in data-dependent acquisition mode with a survey scan resolution of 120 000 (for Orbitrap Fusion) or 60 000 (for Q-Exactive HF) at m/z 200, with an AGC target of 1 x 10e6. Up to 30 of the most intense precursor ions with charge state 2 or higher were sequentially isolated for HCD with normalized collision energy of 27% (Q Exactive HF) or 30% (Orbitrap Fusion), respectively. MS/MS scans were recorded in the Orbitrap for Q Exactive HF, and in the LTQ ion trap for the Orbitrap Fusion. Dynamic exclusion was set to 50 s.

## Data analysis for nucleocytoplasmic partitioning

MS raw files were processed with the MaxQuant software package (version 1.5.0.30) and peak lists were searched with the in-built Andromeda search engine (*Cox and Mann, 2008*; *Cox et al., 2011*).

*FASTA Sequence Databases: for X. laevis* samples an mRNA derived *X. laevis* protein database with 79,214 entries (*Wühr et al., 2014*) was used, for human samples a human UniProt FASTA database (download date: June 2014, 20,258 entries), and for yeast samples a *S. cerevisiae* Uniprot FASTA database (download date: June 2014, 6743 entries). These databases were supplemented with common contaminants (e.g. keratins, serum albumin) and with the reverse sequences of all entries for false discovery rate estimations.

The Andromeda search engine parameters were: carbamidomethylation of cysteine was set as a fixed modification, whereas oxidation of methionine and N-terminal protein acetylation were set as variable modifications; tryptic specificity was considered with proline restriction; up to two missed cleavages were allowed; and the minimum peptide length was set to seven amino acids. The MS survey scan mass tolerance was set to 6 ppm, and MS/MS mass tolerances to 20 ppm (Orbitrap) and 0.5 Da (LTQ ion trap), respectively. The false discovery rate was set to 1% at both the peptide and the protein level.

Several levels of criteria were applied for confident estimation of protein concentrations. First, peptides having posterior error probability (PEP) above 0.01 were excluded from the estimation of protein concentrations in the cytosolic and the nuclear fractions of *X. laevis* samples. Second, proteins identified with single 'only identified by site' peptides in only one compartment were excluded from the calculation of protein concentrations. The absolute protein concentration of proteins in the cytosolic and the nuclear fractions were estimated by correlation with the absolute concentrations of UPS2 standard proteins and their respective iBAQ intensities (*Schwanhäusser et al., 2011*), assuming volumes of 50 nl for the nucleus and 500 nl for the yolk-free cytosol. A regression curve of the absolute amount of UPS2 standard proteins were plotted against their measured iBAQ intensity (log 10 scale) in each biological replicate to generate linear regression equations. These equations were then used to estimate protein concentrations in each biological replicate. Then, the nuclear-to-cytosolic (N:C) ratio was calculated from average protein concentration in the nucleus and the cytosol.

## Data annotation

Annotations were largely based on the UniProt database (*UniProt Consortium, 2015*). For each hit, relevant UniProt data were fetched. These data were simplified to 'simplified localization' (Cytoplasm, Nucleus or Both) and Flags (transmembrane, mitochondrial, ER proteins).

UniProt annotations of *X. laevis* proteins are still sparse. Therefore, human and *X. tropicalis* entries were used as additional references. Appropriate orthologues were mapped by blasting identified *X. laevis* contigs against human and *X. tropicalis* databases (with an E value cut-off of $10^{-11}$).

Functional protein groups were acquired from the Kyoto Encyclopedia of Genes and Genomes (KEGG) database (*Kanehisa et al., 2004*) if not stated otherwise.

For assessment of protein complexes in *Xenopus laevis* oocyte and HeLa cells, human data sets from *Ruepp et al., 2010* and *Havugimana et al., 2012* were used. Yeast protein complexes were assigned according to *Gavin et al., 2006*. For previous CRM1 interaction data, BioGRID (*Chatr-Aryamontri et al., 2015*), the NESdb (*Xu et al., 2012b*), and data from *Thakar et al., 2013* were used.

## Data processing for identification of CRM1 cargoes

Proteins identified with 'only identified by site' (MaxQuant version 15.0.30) within the 'CRM1 +RanGTP' sample were excluded from further analysis. For *X. laevis* samples, these additional parameters were applied: peptides with PEP values higher than 0.01 were excluded, and a minimum of two unique peptides were required to consider a protein for quantification.

For calculating the molar fraction of a given protein in the 'CRM1+RanGTP' sample, its iBAQ intensity was divided by the sum of iBAQ intensities of all proteins detected in this sample.

'Enrichment from input' for a given protein was obtained by dividing its molar fraction within the 'CRM1+RanGTP' sample by its molar fraction in the input extract.

The 'RanGTP-stimulation' for a given protein was calculated by dividing its iBAQ intensities in the 'CRM1+RanGTP' and the 'CRM1 w/o Ran' samples.

When a protein was not identified in the 'CRM1 w/o Ran' or 'Input' sample, then its iBAQ intensity was replaced by a conservative estimate, namely a baseline intensity for detectability (the

median of iBAQ intensities of the least 30 abundant proteins in this sample). This was to avoid in the calculation of parameters divisions by zero and to avoid over-estimating the significance of low abundance cargo candidates.

Based on these three parameters, distinct significance categories were constructed for three species as summarized in the 'Category thresholds' tables (*Supplementary files 2–4*). For proteins that are part of the nuclear transport machinery (Importins, exportins, Nups and related factors) separate categories (NUPs, NTRs, NPC, CRM1 cofactor) were assigned.

## Recombinant protein expression and purification

mmCRM1, scCRM1 and hsRanQ69L$^{5-180}$ were expressed with an N-terminal His14-ZZ-bdSUMO tag and purified by (*i*) immobilisation via $Ni^{2+}$ chelate chromatography, (*ii*) on column bdSENP1 protease elution (*Frey and Görlich, 2014*) and (*iii*) gel filtration. Candidate cargoes and controls were expressed as His14-ZZ-bdSUMO fusions and purified via $Ni^{2+}$ chelate chromatography and imidazole elution.

## Binding assays for validation of direct binding to CRM1

250 pmoles of His14-ZZ-bdSUMO tagged cargo candidates and controls were immobilized on 20 µl anti-ZZ beads, and incubated with 300 pmoles mmCRM1 either in the presence or absence of 1500 pmoles RanGTP (hsRanQ69L$^{5-180}$) in a volume of 500 µl. After 3 hr incubation at 4°C, bound material was eluted by adding bdSENP1 protease, and eluted fractions were analysed by SDS-PAGE and Coomassie-staining.

## Transient HeLa cell transfections and fluorescence microscopy

Cargo and NES candidates were cloned behind either GFP or a GFP-NLS (SV40) module in modified pEGFP-C1 (Invitrogen) vectors. Also a control vector coding for a RFP-NLS-NES fusion was prepared for cotransfection with the GFP/GFP-NLS constructs. HeLa Kyoto cells were grown on coverslips in 24-well plates, and transiently cotransfected with FuGENE6 (Promega) according to manufacturer's instructions. After 24 hr, cells were incubated with either 10 nM Leptomycin B (LMB, dissolved in DMSO) or DMSO alone for 3 hr. They were then fixed for 30 min with 3% paraformaldehyde and 0.1% glutaraldehyde, and the aldehydes were quenched with 1 mg/ml $NaBH_4$. Fluorescence signals were recorded on an SP5 confocal laser scanning microscope (Leica), using sequential scans with excitations at 488 and 561 nm.

## Database depositions

The mass spectrometry proteomics data have been deposited to the ProteomeXchange Consortium (*Vizcaíno et al., 2014*) via the PRIDE partner repository with the dataset identifier PXD002899. The dataset of high confidence CRM1 interactions (cargo categories A, B, and 'low abundance') has been submitted to the IntAct databases (*Orchard et al., 2014*) with the identifier IM-24624.

## Acknowledgements

We wish to thank Volker Cordes for dissecting oocytes, Jens Krull, Anne Weber, Uwe Plessmann and Gabriele Hawlitscheck for excellent technical assistance, the group of Reinhard Lührmann for a generous supply with yeast and HeLa cell extracts, Sinem Saka, Volker Cordes and Connie Paz for critical reading and the Max-Planck-Gesellschaft as well the Deutsche Forschungsgemeinschaft (SFB 860) for funding this work.

## Additional information

### Funding

| Funder | Author |
| --- | --- |
| Max-Planck-Gesellschaft | Koray Kırlı<br>Samir Karaca<br>Heinz Jürgen Dehne<br>Matthias Samwer<br>Kuan Ting Pan<br>Christof Lenz<br>Henning Urlaub<br>Dirk Görlich |
| Deutsche Forschungsgemeinschaft | Henning Urlaub<br>Dirk Görlich |

The funders had no role in study design, data collection and interpretation, or the decision to submit the work for publication.

### Author contributions

KK, SK, Conception and design, Acquisition of data, Analysis and interpretation of data, Drafting or revising the article; HJD, Performed experiments, Acquisition of data; MS, Performed initial experiments, Conception and design; KTP, Setup the system of high pH reverse phase HPLC, Acquisition of data; CL, Assistance in manuscript instrumentation, Acquisition of data; HU, Conception and design, Analysis and interpretation of data; DG, Conception and design, Analysis and interpretation of data, Drafting or revising the article

### Author ORCIDs

Koray Kırlı, http://orcid.org/0000-0002-2289-0652
Samir Karaca, http://orcid.org/0000-0003-4211-9053
Heinz Jürgen Dehne, http://orcid.org/0000-0002-1068-6697
Matthias Samwer, http://orcid.org/0000-0003-4495-8619
Kuan Ting Pan, http://orcid.org/0000-0001-6974-5324
Christof Lenz, http://orcid.org/0000-0002-0946-8166
Henning Urlaub, http://orcid.org/0000-0003-1837-5233
Dirk Görlich, http://orcid.org/0000-0002-4343-5210

### Ethics

Animal experimentation: Our work with Xenopus laevis oocytes has been in accordance with all applicable animal welfare regulations and has been approved by the responsible authority ("Niedersächsisches Landesamt für Verbraucherschutz und Ernährungssicherheit"; file number 33.42502-05/A-005/07).

## Additional files

### Supplementary files

• Supplementary file 1. Nucleocytoplasmic protein partitioning in *Xenopus oocytes*.

• Supplementary file 2. CRM1 binder analysis of Xenopus proteins.

• Supplementary file 3. CRM1 binder analysis of HeLa proteins.

• Supplementary file 4. CRM1 binder analysis of yeast proteins.

• Supplementary file 5. Compares newly identified CRM1 cargoes from human HeLa cells with previous identifications listed in BioGRID, the NESdb database, and by *Thakar et al. (2013)*.

• Supplementary file 6. A fasta file with sequences of proteins that have been tested for a leptomycin B-sensitive localisation in HeLa cells (*Figures 3*, *5–8, and Figure 7—figure supplement 1*)

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
