## [Decision Letter]

Thank you for submitting your work entitled "A deep proteomics perspective on CRM1-mediated nuclear export and nucleocytoplasmic partitioning" for consideration by *eLife*. Your article has been favorably evaluated by Randy Schekman (Senior Editor), Karsten Weis (Reviewing Editor), and two peer reviewers. One of the two reviewers, Naoko Imamoto, has agreed to share her identity.

The reviewers have discussed the reviews with one another and the Reviewing editor has drafted this decision to help you prepare a revised submission.

Summary:

In this manuscript, Kirli et al. describe a thorough and comprehensive identification of CRM1 cargoes in yeast, *Xenopus* oocytes and human cells. Both reviewers felt that this report provides important novel insight on the nucleocytoplasmic distribution of proteins in eukaryotic cells and represents a valuable resource not only to the nuclear transport field, but also to the broad field of cell biology. Furthermore, this study reports several unexpected findings. For example, that a subset of protein kinases is excluded from the nucleus by CRM1 is most likely to ensure compartment-specific phosphorylation. Interestingly, in addition to protein kinases, a broad range of cytoplasmic proteins contain NESs and are bona fide CRM1 cargoes, indicating that eukaryotic cells have evolved a mechanism to maintain these proteins in the cytoplasmic compartment and that the accumulation of these factors in the nuclear compartment can be detrimental.

Essential revisions:

A concern amongst the reviewers was the fact that the purifications were apparently only performed once and no biological replicates are presented in the study. Clearly, there could be significant variation between experiments. This could be particularly problematic for proteins that were not identified, and which do not necessarily need to correspond to low-abundance proteins. For example, the iBAQ intensities vary significantly between experiments (i.e. in one experiment, iBAC intensity is 0, while in the other experiment, iBAQ intensity is high). Repeat experiments (for at least one model system) could clarify this point and potentially strengthen the paper significantly. If this was not possible in a reasonable time frame some of the conclusions need to be toned down and no conclusions should be drawn from the lack of identification/negative results.

Also, it was unclear how the thresholding was performed. On one hand, e.g. in [Supplementary-material SD2-data] (*Xenopus* CRM1 cargo, INDEX), "RanGTP stimulation" was above 500, "Input enrichment" was above 4, "CRM%" was above 0.005 for A1 categories. On the other hand, "Input enrichment" was above 100, "CRM1% " was above 0.005% for A2 categories, and so on. More explanation is required and it is not clear how these lines were drawn and the thresholds were set.

---

## [Author Response]

Essential revisions:

A concern amongst the reviewers was the fact that the purifications were apparently only performed once and no biological replicates are presented in the study. Clearly, there could be significant variation between experiments. This could be particularly problematic for proteins that were not identified, and which do not necessarily need to correspond to low-abundance proteins. For example, the iBAQ intensities vary significantly between experiments (i.e. in one experiment, iBAC intensity is 0, while in the other experiment, iBAQ intensity is high). Repeat experiments (for at least one model system) could clarify this point and potentially strengthen the paper significantly. If this was not possible in a reasonable time frame some of the conclusions need to be toned down and no conclusions should be drawn from the lack of identification/negative results.

The analysis of nucleocytoplasmic partitioning did include three biological replicates. These were performed using different biological samples (note that there is also genetic variability between frogs) by different workflows (SDS-PAGE⇒ tryptic digests ⇒ LC/MS versus in-solution-digest ⇒ two dimensional HPLC⇒ MS) and different mass spectrometric setups. Yet, the agreement in terms of identification is remarkably good: Out of the 1000 top abundant proteins identified in the first replicate, 996 were re-identified in the second, and all 1000 in the third (see table below). For low abundance proteins, the re-discovery was, of course, lower.

Re-identification of proteins from Biol. Repl. 1Test set of proteins identified in Biological Replicate 1 from *Xenopus* oocyte nuclei, sorted by abundance:in Biol. Repl. 2in Biol. Repl. 3Top 100099.6%100.0%1001-200196.3%98.1%Least 100087.7%71.8%

The analysis of CRM1-cargoes was performed with just two technical replicates each. A biological replicate of all binding experiments at the already presented depth would require 4 weeks of instrumentation time and this cannot be accommodated within the time frame of the revision, also out of respect for other projects that rely on the instruments as well.

Our text did not include any specific conclusions from negative results – other than the suggestion that the extremes (category A cargoes and clear non-binders) could provide a statistical basis for optimising NES-predictions. We still believe this to be a valid point.

Yet, we are also fully aware of the fact that any large-scale proteomics study is likely to produce false-positive, as well as false-negative, hits. We therefore decided to look deeper into this issue.

First, we implemented a more conservative classification of cargoes as well as of ‘CRM1 non-binders’. Essentially, candidates are listed within these categories only if their abundance exceeds a certain threshold in either the ‘CRM1+RanGTP’-bound fraction or in the starting material. For cargoes, the threshold became stricter (the so far least reliable ‘category D’ became a subgroup of ambiguous). For non-binders, a threshold for ‘input abundance’ was newly implemented. This reduced the numbers of non-binders to the most reliable ones, while the category ‘ambiguous’ became larger.

Second, we used an earlier analysis of CRM1-bound fractions from HeLa cell extracts as a ‘biological replicate 2’ in order to validate the main dataset (biological replicate 1). While still optimising the cargo identification, we used the SILAC (stable isotope labeling by amino acids in cell culture) approach to quantitatively compare ‘CRM1+RanGTP’-bound fractions with a minus RanGTP control (at that time we had not yet fully realized that a comparison of just those two sample does not allow a reliable identification of cargoes). We now re-analysed the SILAC data, derived iBAQ intensities and calculated molar fractions for 2756 identified proteins in the ‘CRM1+RanGTP’ sample.

Figure 10 shows how the measured molar fractions correlate between the two datasets. Considering that different instrumentations were used, the Pearson correlation coefficient of 0.79 is actually quite good and in full agreement with our estimation that protein quantitation with our setup is, on average, accurate within a factor of 2.3.

Author response image 1.Scatter plot shows correlation of molar fractions of given protein species within the ‘CRM1+RanGTP’-bound fractions of the two biological replicates.Note that many of the data points overlap at the diagonal. The local density of data points in the plot was therefore color-coded like a heat map, with red indicating the highest density and blue the lowest density. The red-orange region represents already 50% of the data points.**DOI:**
http://dx.doi.org/10.7554/eLife.11466.020

Third, the correlation plot does not include proteins that remained undetected in the ‘CRM1+RanGTP’-bound fractions and thus excludes many of the ‘CRM1 non-binders’. We therefore compared the two datasets in yet another way: We re-calculated the cargo classification after replacing the iBAQ intensities in the ‘CRM1+RanGTP’-bound fraction by the (re-scaled) iBAQ intensities of biological replicate 2. […] The agreement between the replicates is remarkably good. Most assignments (97%) are identical. There are no swaps between cargo and non-binders, just a few changes between ambiguous/cargo and ambiguous/non-binder.

Fourth, we added new the Figure 7—figure supplement 1, where we validated predicted negatives from our screen for CRM1 cargoes by expressing them as GFP or GFP-NLS fusions in HeLa cells. Specifically, we tested seven (out of the eleven) cases of clearly conflicting data, namely human proteins that are listed as CRM1 cargoes in the NESdb, but appeared negative according to our proteomics data (the seven were the ones, where cloning from HeLa cDNA worked at the first trial). Four of them showed an exclusively nuclear signal already as plain GFP fusions. The other three were equally distributed between nucleus and cytoplasm, and their nuclear signal did not increase upon blocking CRM1 by leptomycin B. These seven re-tested candidates thus behaved as predicted from their classification as CRM1 non-binders, at least for the tested protein isoforms (cloned from HeLa cell cDNA), for the tested cell type (human HeLa cells) and under standard cell culture conditions.

Also, it was unclear how the thresholding was performed. On one hand, e.g. in [Supplementary-material SD2-data] (Xenopus CRM1 cargo, INDEX), "RanGTP stimulation" was above 500, "Input enrichment" was above 4, "CRM%" was above 0.005 for A1 categories. On the other hand, "Input enrichment" was above 100, "CRM1% " was above 0.005% for A2 categories, and so on. More explanation is required and it is not clear how these lines were drawn and the thresholds were set.

We agree, the ‘explanatory tables’ for the thresholds were confusing. In fact, they were a legacy from an earlier version. This has been corrected now and the layout has been simplified.